# Loss of neuropeptidergic regulation of cholinergic transmission induces homeostatic compensation in muscle cells to preserve synaptic strength

**Jiajie Shao[1,2], Jana F. Liewald[1,2], Wagner Steuer Costa[1,2], Christiane Ruse[1,2], Jens Gruber[1,2], Mohammad S. Djamshedzad[1,2], Wulf Gebhardt[1,2], Alexander Gottschalk** [1,2]*

1 Faculty of Molecular Sciences, Institute of Biophysical Chemistry, Goethe University, Frankfurt, Germany, 2 Buchmann Institute for Molecular Life Sciences, Goethe University, Frankfurt, Germany

* a.gottschalk@em.uni-frankfurt.de

## Abstract

Chemical synaptic transmission at the neuromuscular junction (NMJ) is regulated by electrical activity of the motor circuit, but may also be affected by neuromodulation. Here, we assessed the role of neuropeptide signaling in the plasticity of NMJ function in *Caenorhabditis elegans*. We show that the CAPS (Ca$^{2+}$-dependent activator protein for secretion) ortholog UNC-31, which regulates exocytosis of dense core vesicles, affects both pre- and post-synaptic functional properties, as well as NMJ-mediated locomotion. Despite reduced evoked acetylcholine (ACh) transmission, the loss of *unc-31* results in a more vigorous response to presynaptic stimulation, i.e., enhanced muscle contraction and Ca$^{2+}$ transients. Based on expression profiles, we identified neuropeptides involved in both cholinergic (FLP-6, FLP-15, NLP-9, NLP-15, NLP-21, and NLP-38) and GABAergic motor neurons (FLP-15, NLP-15), that mediate normal transmission at the NMJ. In the absence of these peptides, neurons fail to upregulate their ACh output in response to increased cAMP signaling; for *flp-15; nlp-15* double mutants, we observed overall increased postsynaptic currents, indicating that these neuropeptides may be inhibitory. We also identified proprotein convertases encoded by *aex-5*/*kpc-3* and *egl-3*/*kpc-2* that act synergistically to generate these neuropeptides. We propose that postsynaptic homeostatic scaling, mediated by increased muscle activation, likely through excitability, might compensate for the reduced cholinergic transmission in mutants affected for neuropeptide signaling, thus maintaining net synaptic strength. We show that in the absence of UNC-31 muscle excitability is modulated by upregulating the expression of the muscular L-type voltage-gated Ca$^{2+}$ channel EGL-19. Our results unveil a role for neuropeptidergic regulation in synaptic plasticity, linking changes in presynaptic transmission to compensatory changes in muscle excitability.

**Data availability statement:** All relevant data are presented within the figures and supplemental figures of the paper and numerical data are provided in the Supporting S1 Data to S9 Data for each main figure and in S1 Dataset for the supplemental figures, with individual tabs for each supplementary figure.

**Funding:** This work was funded by the Deutsche Forschungsgemeinschaft (DFG), CRC1080 project B02 (https://gepris.dfg.de/gepris/projekt/227591453), and by core support from Goethe University ((https://www.uni-frankfurt.de/49115505), both to A.G. C.R. is a fellow (Kekulé stipend, K 215/54; https://www.vci.de/fonds/stipendien/kekule-stipendium/seiten.jsp) of the Fonds der Chemischen Industrie (FCI). The funders had no influence on the content of the research.

**Competing interests:** The authors have declared that no competing interests exist.

**Abbreviations :** ACh, acetylcholine; AChE, ACh esterase; AID, auxin-induced degradation; APs, action potentials; DCV, dense core vesicle; IPTG, isopropyl-β-D-thiogalacto-pyranosid; MNs, motor neurons; mPSCs, miniature postsynaptic current; MWT, multiworm tracker; nAChRs, nicotinic acetylcholine receptors; NGM, nematode growth media; NMJ, neuromuscular junction; PCs, proprotein convertases; PSCs, postsynaptic currents; SVs, synaptic vesicles; vAChT, vesicular ACh transporter; VGCCs, voltage-gated $Ca^{2+}$ channels; wt, wild-type.

## Introduction

Chemical synaptic transmission at NMJs is orchestrated by networks of motor neurons (MNs) and pre-motor interneurons, and involves pattern generation such that coordinated movement and also locomotion can occur [1,2]. The patterns of MN electrical activity lead to the fusion of synaptic vesicles (SVs) and release of neurotransmitter [3]. The rate of SV fusions depends on the extent and duration of depolarization of the synaptic terminal, which triggers voltage-gated $Ca^{2+}$ channels (VGCCs) [4]. It also depends on the mobilization of SVs from the reserve pool, such that depolarization can cause SV fusion at a higher rate. These mechanisms enable plasticity of the regulation of synaptic transmission, and also enable presynaptic homeostatic upscaling [4–10]. This can occur in situations where transmitter output is abnormal, or in response to (reduced or missing) retrograde signaling from the postsynaptic side, when neurotransmitter detection is compromised. This is well characterized for the NMJ of *Drosophila* [11–13]. There are also mechanisms that involve postsynaptic homeostatic scaling in response to lower-than-normal presynaptic transmitter release. E.g., chronically blocking presynaptic firing using tetrodotoxin, or increasing it by GABA blockers, caused an increase or decrease in miniature postsynaptic current (mPSCs) amplitude, respectively [14]. However, there may also be other postsynaptic mechanisms in response to a chronic reduction of presynaptic transmitter release.

The NMJ of the nematode *Caenorhabditis elegans* comprises cholinergic and GABAergic MNs that directly innervate muscle. Moreover, cholinergic neurons that innervate muscles ventrally also synapse onto GABAergic neurons that innervate dorsal muscle, in order to enable body bending to occur; this topology also occurs in an inverted fashion, by different subsets of MNs [15]. GABAergic feedback to cholinergic neurons is mediated by $GABA_B$ receptors [16,17]. Optogenetic stimulation experiments suggest that cholinergic MNs desensitize in response to repeated or prolonged stimulation, while GABAergic neurons appear to be potentiated [16,18–20], thus demonstrating acute presynaptic plasticity. An additional layer of regulation occurs through neuropeptides, which were released in response to optogenetic cAMP generation and, in an autocrine fashion, caused a modulation of the ACh content of SVs [21]. Another publication described the regulation of presynaptic cholinergic output when the function of ACh esterase (AChE) is inhibited, and this regulation involved neuropeptides that were released from a proprioceptive neuron, DVA [22]. Furthermore, postsynaptic feedback mechanisms cause regulation of presynaptic transmission. One such mechanism is mediated through a microRNA (encoded by *mir-1*) that regulates postsynaptic nAChR expression but also presynaptic ACh release; the latter involves the extracellular cleavage of postsynaptic neurexin, that is released from the muscle surface in response to excess ACh signaling and inhibits presynaptic CaV2 channels and reduces abundance of the SV protein RAB-3 [23–25]. How muscle responds to altered ACh signaling (in this case, increased cholinergic signaling) to induce these changes is not well understood, but involves regulation of the MEF-2 transcription factor [23]. Whether muscle responds in a plastic way to reduced presynaptic ACh signaling is unknown.

cAMP signaling induced by optogenetics, using the photoactivated adenylyl cyclase bPAC in cholinergic neurons, causes an increase in the rate of SV fusion, but also enhances neuropeptide release [21]. A major mode of action of these neuropeptides is to presynaptically promote filling of ACh into SVs via the vAChT (encoded by *unc-17*). This was evident at the level of mPSCs, which exhibited increased mPSC rate and amplitude, and by the size of SVs as observed by electron microscopy, showing that bPAC stimulation rapidly induces an increase of SV diameter. In *unc-31* mutants, lacking CAPS, that is required for dense core vesicle (DCV) fusion and in which the release of neuropeptides is largely abolished, the increase of mPSC amplitudes was absent, and these animals had smaller SV diameters. Synaptic transmission at the NMJ is largely reduced in *unc-31* mutants, as both electrically or optogenetically evoked ePSCs are statistically significantly smaller compared to wild type [26,27]. This is in line with the idea that SVs in these animals contain less ACh. The identity of the neuropeptides mediating this regulation at the NMJ is unclear, though NLP-12 neuropeptides, released from the proprioceptive neuron DVA [28], were previously shown to affect cholinergic signaling at the NMJ, and to contribute to presynaptic plasticity [22]. Animals that had been treated with the AChE inhibitor aldicarb showed homeostatic upregulation of synaptic transmission that depended on the neuropeptide receptor CKR-2, expressed in cholinergic neurons. In recent years, single-cell RNAseq data identified the neuropeptides and neuropeptide receptors expressed in the MNs of *C. elegans*, as well as a functional analysis of cognate neuropeptide/neuropeptide receptor pairs [29–32].

Here, we functionally probed candidate neuropeptides and analyzed how the lack of neuropeptide signaling affects the NMJ. We identified several neuropeptides expressed in cholinergic and GABAergic MNs as being involved in cAMP-modulated cholinergic transmission, using behavioral assays and candidate mutants. NLP-9 and NLP-38 are potentiators of cholinergic mPSC amplitude, while FLP-15 and NLP-15 convey inhibition, from GABAergic and, possibly, cholinergic neurons. We show an involvement of other neuropeptides (and neuropeptide receptors) that depend on processing by two of the four proprotein convertases (PCs), AEX-5 and EGL-3, of which AEX-5 is specifically required in GABAergic neurons. Lack of these neuropeptides causes postsynaptic homeostatic upscaling of the ACh response in muscle. This was not affected through regulation of ACh receptors, but by increasing expression of the L-type VGCC EGL-19/CaV1. Lack of neuropeptide signaling induced generation of larger muscle action potentials (APs), thus muscles upregulate their excitability to balance the reduced presynaptic ACh release.

## Results

### UNC-31/CAPS is required in cholinergic and GABAergic neurons for normal synaptic transmission at the NMJ

To identify neuropeptides functionally involved in cAMP regulation of transmission at the NMJ, we first tested different approaches. We used behavioral assays, combined with optogenetic bPAC stimulation of the cholinergic MNs, in wild-type (wt) and *unc-31(n1304)* mutant animals. First, we assayed crawling locomotion speed [33], second, we measured body length as a readout for muscle tone [18], and third, we quantified $Ca^{2+}$ transients in postsynaptic muscle, using the genetically encoded fluorescent $Ca^{2+}$ indicator RCaMP [34] (Fig 1A–1F). Acutely activating bPAC in cholinergic MNs induced a significant speed increase in crawling animals [21] (Fig 1A and 1B). The speed increase observed in wt animals was robust, but significantly larger in *unc-31* (null alleles *n1304* and *e928* essentially had the same phenotype). bPAC expression was not affected in *unc-31* mutants, as judged by bPAC::YFP signal in ventral cord cholinergic MNs (S1A and S1B Fig). bPAC activation was further accompanied by mild body contraction (Fig 1C and 1D), likely due to the coordinated activity of body wall muscles (BWMs) during locomotion induction. In this assay, *unc-31* mutants showed much reduced basal locomotion speed, as previously observed [35], likely due to the general defect in neuropeptide signaling throughout the nervous system. However, this phenotype may also be more specifically due to reduced evoked ACh release at the NMJ of *unc-31* mutants [26,27] (S1C and S1D Fig). Thus, we expected a smaller increase of locomotion speed in response to bPAC activation in *unc-31* animals. Unexpectedly, however, they displayed a much more pronounced speed increase (almost reaching absolute speed levels of wt animals; Fig 1A and 1B), as well as larger accompanying muscle contraction (Fig 1C and 1D). This extent of the speed increase was surprising, given the fact that *unc-31* animals

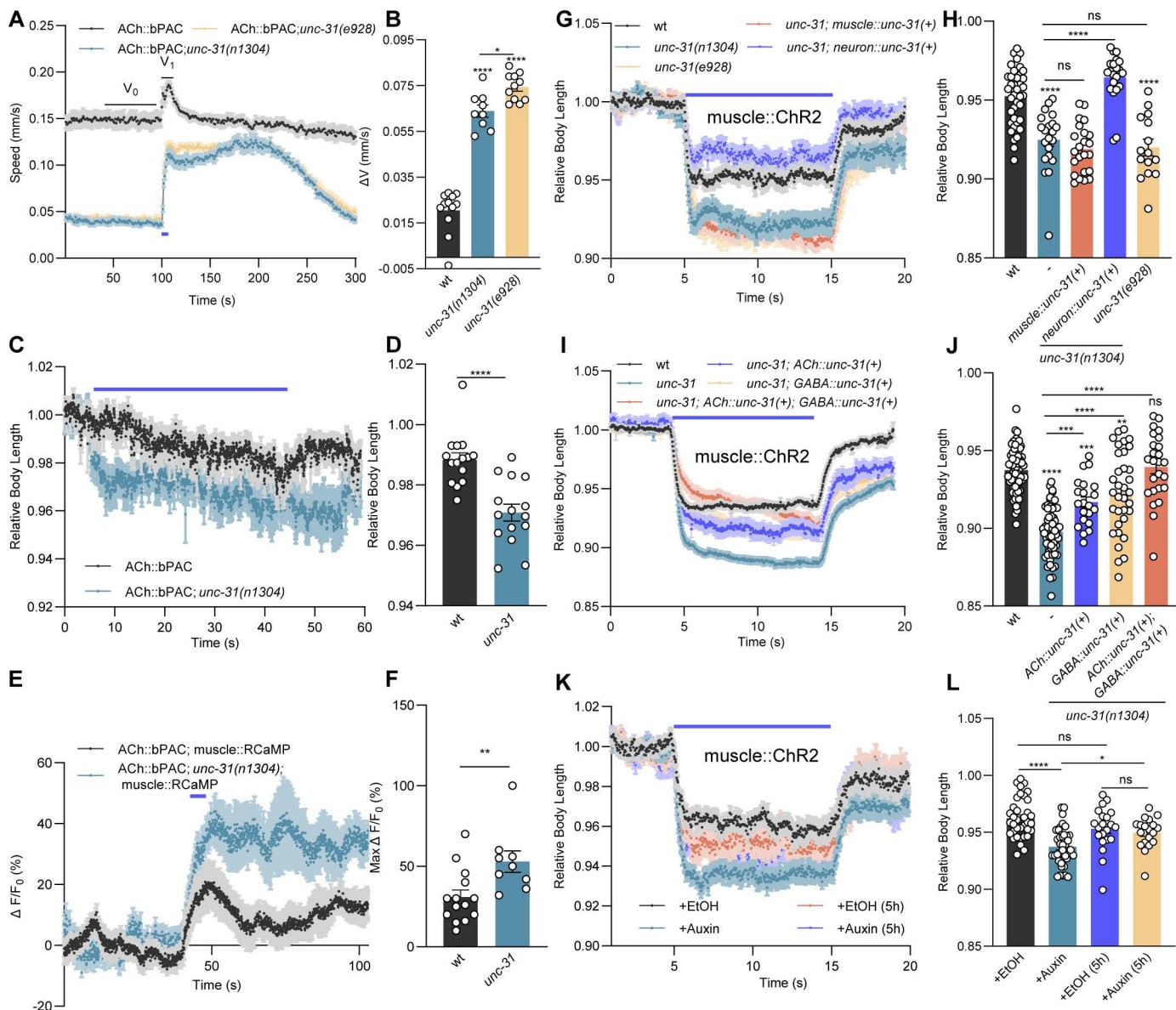

**Fig 1. UNC-31/CAPS functions in cholinergic and GABAergic neurons to regulate synaptic transmission at the NMJ. (A, B)** Measurements of crawling speed of bPAC transgenic animals (using the unc-17 promoter for cholinergic expression) on a multiworm tracker (MWT) device [33]. Blue bar indicates 5 s period of blue light illumination. The average speed from 50 to 100 s was taken as basal speed $V_0$ and from 100 to 120 s as $V_1$. $\Delta V$, as shown in panel B, indicates the difference between $V_0$ and $V_1$. wt ($n = 60–80$, $N = 12$), unc-31 (n1304) ($n = 60–80$, $N = 9$), and unc-31(e928) mutants ($n = 60–80$, $N = 11$) were compared. An intensity of 70 μW/mm² blue light was used for bPAC stimulation throughout this study. **(C, D)** Measurements of (induced changes in) body length along the midline of single animals, before, during, and after bPAC activation in cholinergic neurons. Absolute length values of worms (in pixels) were normalized to the averaged values before illumination (0–4 s, normalization was done for each animal) to obtain the presented relative body length. The data in panel D represent the mean values over the entire illumination period (5–45 s) shown in panel C. Blue bar indicates blue light illumination from 5 to 45 s. (wt, $n = 15$; unc-31, $n = 15$). **(E, F)** Calcium imaging using the genetically encoded fluorescent indicator RCaMP in body wall muscle cells, increase evoked by bPAC stimulation of cholinergic neurons. RCaMP signal is normalized as $\Delta F/F_0$, where $F_0$ means the average signal of the first 40 s before blue light illumination, $\Delta F$ indicates the difference between the fluorescent signals $F$ and $F_0$. Max $\Delta F/F_0$ denotes the maximal signal observed after bPAC stimulation. Blue bar indicates blue light illumination from 45 to 50 s. (wt, $n = 14$; unc-31, $n = 9$). **(G–J)** Measurements of body length induced by muscular ChR2 activation using 0.1 mW/mm² blue light stimulation. unc-31 rescue was performed by specifically expressing UNC-31 in muscles or neurons of unc-31(n1304) mutants, respectively (G and H; animal number tested in each group: $n = 38, 21, 23, 21, 15$, from left to right, respectively), as well as in cholinergic or GABAergic neurons, and simultaneously in both neural types, respectively (I and J; animal number tested in each group: $n = 62, 58, 21, 34, 25$, from left to right). Blue bar indicates the 5–15 s blue light illumination. **(K, L)** Measurements of body

length induced by muscular ChR2 activation after UNC-31 depletion using 0.065 mW/mm² blue light stimulation. Auxin- or ethanol-treated animals were compared. Animal number tested in each group: $n = 36, 40, 21, 19$, from left to right. Blue bar indicates the 5–15 s blue light illumination. The data in panels H, J, and L represent the mean values over the entire illumination period (5–15 s) shown in panels G, I, and K, respectively. All data are presented as mean ± SEM. Statistical significance for two-group datasets and multiple-group datasets comparison was determined using unpaired $t$ test and one-way ANOVA with Tukey correction, respectively. *, **, ***, and **** indicate $p < 0.05$, $p < 0.01$, $p < 0.001$, and $p < 0.0001$, respectively. Numerical data can be found in S1 Data.

have much smaller evoked currents and smaller mPSCs (S1C and S1D Fig). Possibly, compensatory changes increase the response of muscle to the reduced cholinergic transmission, a phenotype we observed earlier for mutants affecting presynaptic transmitter release [18]. In line with the behavioral results, presynaptic bPAC photoactivation elicited a more robust increase in muscle Ca²⁺ levels, as measured by RCaMP in *unc-31* mutants, compared to wt (Fig 1E and 1F). This indicates that reduced ACh signaling causes an increase in muscle Ca²⁺ influx. As a fourth assay, and to test whether postsynaptic muscles may contract more, possibly *via* increased excitability [18], we used direct optogenetic depolarization via channelrhodopsin-2 (ChR2) expressed in BWMs (*pmyo-3* promoter; expression of ChR2 was not affected by *unc-31* mutations; S1E and S1F Fig). Wild type animals showed body contraction by about 5% of the body length in response to photostimulation (Fig 1G and 1H). Indeed, *unc-31* mutants (*e1304* as well as *e928*) showed statistically significantly enhanced body contraction of about 7.5%. To verify this was due to the loss of neuropeptide release from neurons, we rescued UNC-31 in BWMs, or pan-neuronally (*prab-3* promoter) [35]. As expected, UNC-31 function was rescued in neurons, but not in muscle (Fig 1G and 1H).

Neuropeptide signaling affects ACh release at the NMJ [21]. Since UNC-31 affects the release of many if not most neuropeptides, and as different cholinergic and GABAergic MN classes at the NMJ feedback on each other [2,15–17], the source of the specific neuropeptides affecting cholinergic signaling may not be the cholinergic neurons. We thus determined the site of action of UNC-31 in this context. *unc-31* is primarily expressed in the ventral, dorsal, and sublateral nerve cords [35,36]. UNC-31 expression in cholinergic and GABAergic neurons could partially rescue the enhanced body contraction, but only when it was expressed in both cell types, a full rescue was observed (Fig 1I and 1J). These results suggest that both cholinergic and GABAergic neuropeptides contribute to normal signaling at the NMJ.

To confirm the necessity of UNC-31 for regulation of muscle contraction, we selectively depleted it using auxin-induced degradation (AID) [27]. Worms expressing ChR2 in muscles and continuously treated with auxin showed a relative body length similar to animals treated with ethanol, the diluent for auxin, at a light intensity of 0.1 mW/mm²: ethanol: 0.944 versus auxin: 0.945 (S1I and S1J Fig). UNC-31 depletion may have been insufficient, resulting in only a modest increase in contractions as compared to *unc-31* mutants. Thus, any difference may have been masked due to the intense stimulation. Therefore, we applied a milder light stimulus (0.065 mW/mm²). Now, auxin treatment led to a statistically significantly enhanced muscle contraction induced by ChR2, compared to controls (Fig 1K and 1L). Thus, loss of *unc-31* in cholinergic neurons enhances contractions, possibly through effects on postsynaptic excitability. The observed effects could either be caused during development as a response to presynaptic changes, or it could be a more acute effect of altered synaptic signaling, that would also arise post-developmentally. To distinguish these possibilities, we acutely applied auxin or ethanol to adult worms for 5 h. UNC-31 levels were statistically significantly reduced by auxin, but not by ethanol exposure (S1G and S1H Fig), yet the ChR2-induced contraction was normal (Fig 1K and 1L). This suggested that muscle excitability was not altered in adults after UNC-31 depletion. Collectively, these results demonstrate that UNC-31 must be absent during development, or for >5 h, in cholinergic neurons to induce enhanced muscle contractions.

## AEX-5 and EGL-3 proprotein convertases are required for NMJ neuropeptide signaling

Since UNC-31/CAPS is required for exocytosis of DCVs, which can contain different classes of neuropeptides as well as monoamines [36–38], we asked if different neuropeptide subclasses may be involved. The requirement of different

neuropeptide precursor processing enzymes in different cell types may provide clues as to which neuropeptide(s) may be involved in NMJ regulation. Neuropeptide precursors are processed by PCs and carboxypeptidases to achieve mature and active neuropeptides [39–41]. Thus, we compared the evoked muscle contractions of wt and mutants of the four known *C. elegans* PCs. Of those, *aex-5/kpc-3*, *bli-4/kpc-4*, and *kpc-1,* encode homologs of human PC1 with a cleavage sequence R-X-X-R, while the PC2 homolog EGL-3/KPC-2 is specific for cleavage following RR or KR sequences [42] (S1 Table). The *aex-5/kpc-3* mutants exhibited statistically significantly enhanced muscle contraction compared to wt during ChR2 activation of BWMs, whereas *egl-3/kpc-2* mutants showed a slight, non-significant increase, and *kpc-1* and *bli-4/kpc-4* mutants showed unaltered contractions (Fig 2A and 2B). *aex-5* mutants, however, were not as compromised as *unc-31* mutants in the same assay. With milder stimulation, *aex-5* mutants also exhibited enhanced evoked muscle contractions (S2A and S2B Fig), that could be rescued by restoring *aex-5* expression from its own promoter. We investigated whether *aex-5* acts

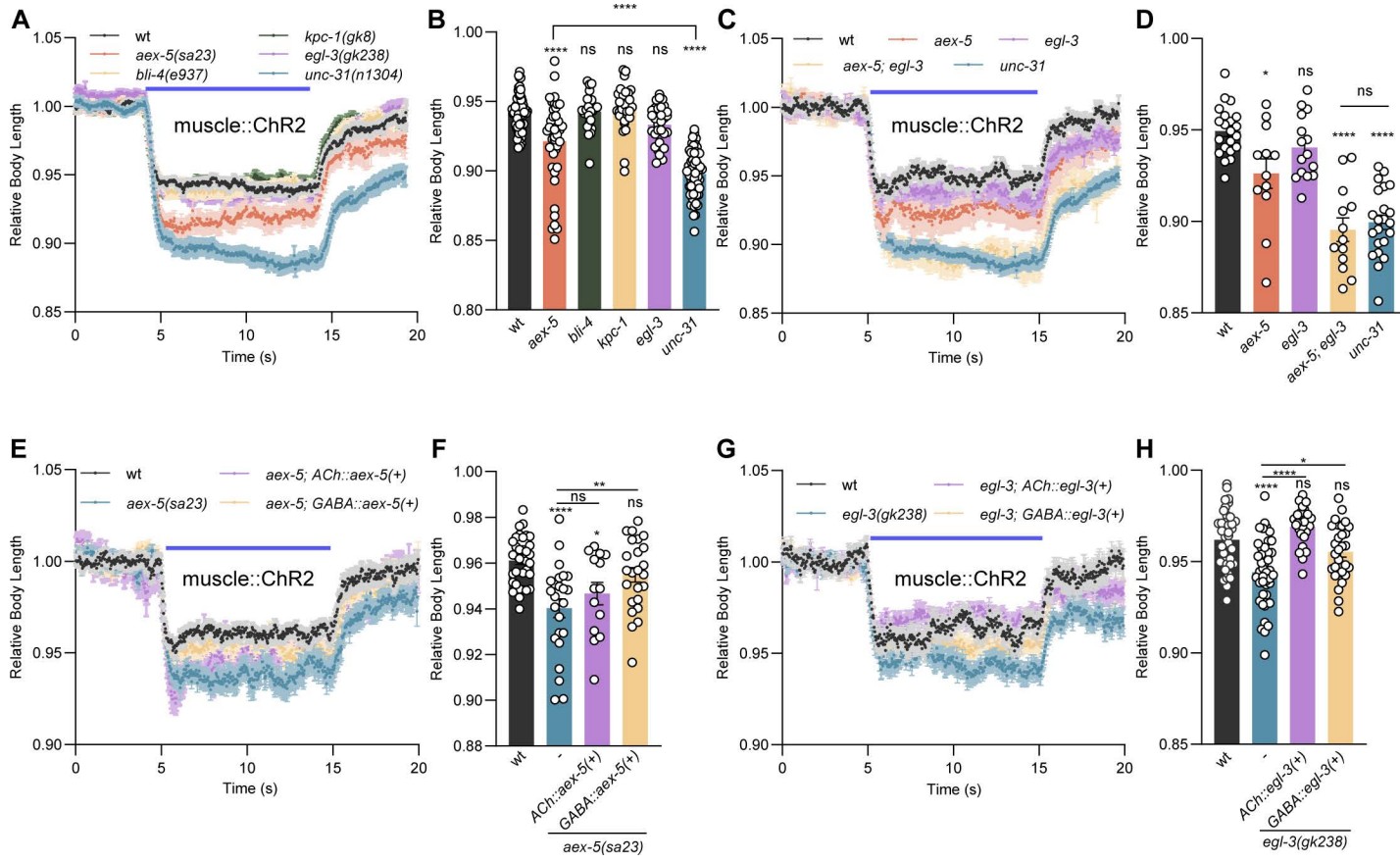

**Fig 2. Pro-protein convertases AEX-5 and EGL-3 are required for neuropeptidergic signaling at the NMJ. (A, B)** Measurements of body length induced by muscular ChR2 activation using 0.1 mW/mm$^2$ blue light stimulation. Mean ± SEM and group data for time period during illumination. Single mutants *aex-5(sa23)* (*n* = 41), *bli-4(e937)* (*n* = 23), *kpc-1(gk8)* (*n* = 31), *egl-3(gk238)* (*n* = 32), and *unc-31(n1304)* (*n* = 58) were compared to wt (*n* = 60). **(C, D)** as in (A, B) for mutants *aex-5* (*n* = 12), *egl-3* (*n* = 15), *unc-31* (*n* = 22), and *aex-5; egl-3* (*n* = 13) doubles, were compared to wt (*n* = 21). Note that the same wt control data was used as in panel A. **(E–H)** Measurements of body length induced by muscular ChR2 activation using 65 μW/mm$^2$ blue light stimulation. *aex-5* (E, F) and *egl-3* (G, H) rescue were performed by specifically expressing AEX-5 and EGL-3 in cholinergic and GABAergic neurons respectively. Animal number tested in E, F: *n* = 35, 24, 14, 24, and in G, H: *n* = 40, 45, 27, 29 from left to right, respectively. In each panel, the blue bar indicates the 5–15 s blue light illumination. The data in panels B, D, F, and H represent the mean values over the entire illumination period shown in panels A, C, E, and G, respectively. All data are presented as mean ± SEM. Statistical significance for multiple-group datasets comparison was determined using one-way ANOVA with Tukey correction. *, **, and **** indicate *p* < 0.05, *p* < 0.01, and *p* < 0.0001, respectively. Numerical data can be found in S2 Data.

synergistically with *egl-3* in regulating NMJ signaling. *aex-5; egl-3* double mutants showed an additive effect, with body contraction that fully recapitulated the *unc-31* mutant phenotype (Fig 2C and 2D), indicating that both enzymes jointly process a set of neuropeptides that are required for normal NMJ signaling, with *aex-5* playing a major role.

AEX-5 processes a peptide in the intestine that is released to regulate the defecation motor program [39,43,44]. AEX-5 has also been reported to regulate AEX-1-dependent retrograde signaling at NMJs [45], fat metabolism [46], and pathogen avoidance [47]. We determined if AEX-5 also functions in cholinergic and GABAergic neurons. The muscle contraction phenotype of *aex-5* mutants was rescued by restoring AEX-5 expression in GABAergic, but not in cholinergic neurons (Fig 2E and 2F). Furthermore, we detected the secretion of AEX-5 from GABAergic neurons, as it could be observed in coelomocytes, scavenger cells that capture released neuropeptides and PCs via bulk endocytosis [44,48] (S2C Fig). These findings demonstrate that not only cholinergic but also AEX-5-dependent GABAergic neuropeptide signaling is involved in regulating NMJ function. We further analyzed in which cell type EGL-3 function was required. This showed that expression of EGL-3 in both cholinergic and GABAergic neurons recovered wt phenotypes, while rescue in cholinergic neurons appeared more effective (Fig 2G and 2H). Thus, neuropeptides affecting NMJ signaling and depending on EGL-3 are likely present in both, cholinergic and GABAergic MNs.

## Functional screen for neuropeptides acting at the NMJ

We next wanted to identify neuropeptides affecting cholinergic signaling and/or the postsynaptic response to ACh release. Due to the large effect size, we chose the locomotion speed assay (bPAC in cholinergic neurons, probing for neuron output) for this purpose. We first tested if absence of neuropeptide subclasses would also be detected, in *egl-3*, *aex-5*, and respective double mutants, and compared them to *unc-31* alleles (Fig 3A and 3B). The most pronounced effect was observed for *aex-5; egl-3* double mutants. *egl-3* mutants were as affected as both tested *unc-31* alleles, while *aex-5* had a milder phenotype, yet was also exhibiting a significantly larger speed increase than wt animals. Next, we addressed individual neuropeptide mutants, or combinations of those, as well as some neuropeptide receptors. Likely expression patterns, based on mRNA, not the actual (pro-)proteins, were extracted from a scRNAseq database [29,30] (S3A Fig). Based on this data, we chose the neuropeptides with the most abundant mRNAs in cholinergic MNs, i.e., FLP-6 (though expressed only in the VC neurons), FLP-9, FLP-18, NLP-9, NLP-13, NLP-15, NLP-21, and NLP-38. We also included neuropeptides expressed in GABAergic neurons that may likewise affect NMJ signaling and/or excitation/inhibition balance, i.e., FLP-15, but also FLP-9 and NLP-15, the latter are also expressed in cholinergic neurons. Further, we included NLP-12, expressed in the proprioceptive tail neuron DVA, as this was previously shown to affect cholinergic MNs and affects body bending; DVA itself is also cholinergic [22,28]. Interestingly, FLP-15 is also highly abundant in DVA. Last, we included receptors for some of these neuropeptides, as indicated in brackets: DMSR-7 (numerous FLP neuropeptides), NPR-18 (NLP-9), SPRR-1 (NLP-42, as a negative control), and SPRR-2 (NLP-38) [31,32,49,50]. We tested these animals for basal locomotion and in response to bPAC activation (Figs 3C, 3D, and S3B). *unc-31* mutants exhibit very slow locomotion. Likewise, nine of the mutants showed locomotion significantly slower than wt, while *flp-9* mutants were faster, and the remaining seven mutants were not different to wt (Fig 3C). Next, we assessed the speed increase following bPAC photoactivation (Figs 3D and S3B). Eleven mutants and the two tested *unc-31* alleles showed a (statistically) significantly higher speed increase compared to wt, while six mutants were not significantly different in their speed increase. The speed increase was very strong for *unc-31* mutants, and also *flp-6*, *flp-15*, *nlp-9*, *nlp-12*, *nlp-15*, *nlp-21*, *nlp-38*, as well as several double or triple mutants showed a speed increase that was statistically significantly more pronounced than in wt (Figs 3D and S3B). This also included the NPR-18 receptor for the NLP-9 neuropeptide, but none of the other neuropeptide receptors. Of the neuropeptides highly expressed in GABAergic neurons, *flp-15,* and *nlp-15, as well as flp-15; nlp-15* double mutants exhibited enhanced speed increase (Figs 3D and S3B), suggesting that the two neuropeptides contribute to regulation of NMJ transmission. In sum, the lack of several of the neuropeptides we tested partially phenocopied *unc-31* mutants. Possibly, combining all of these neuropeptide mutants might fully recapitulate the *unc-31* phenotype.

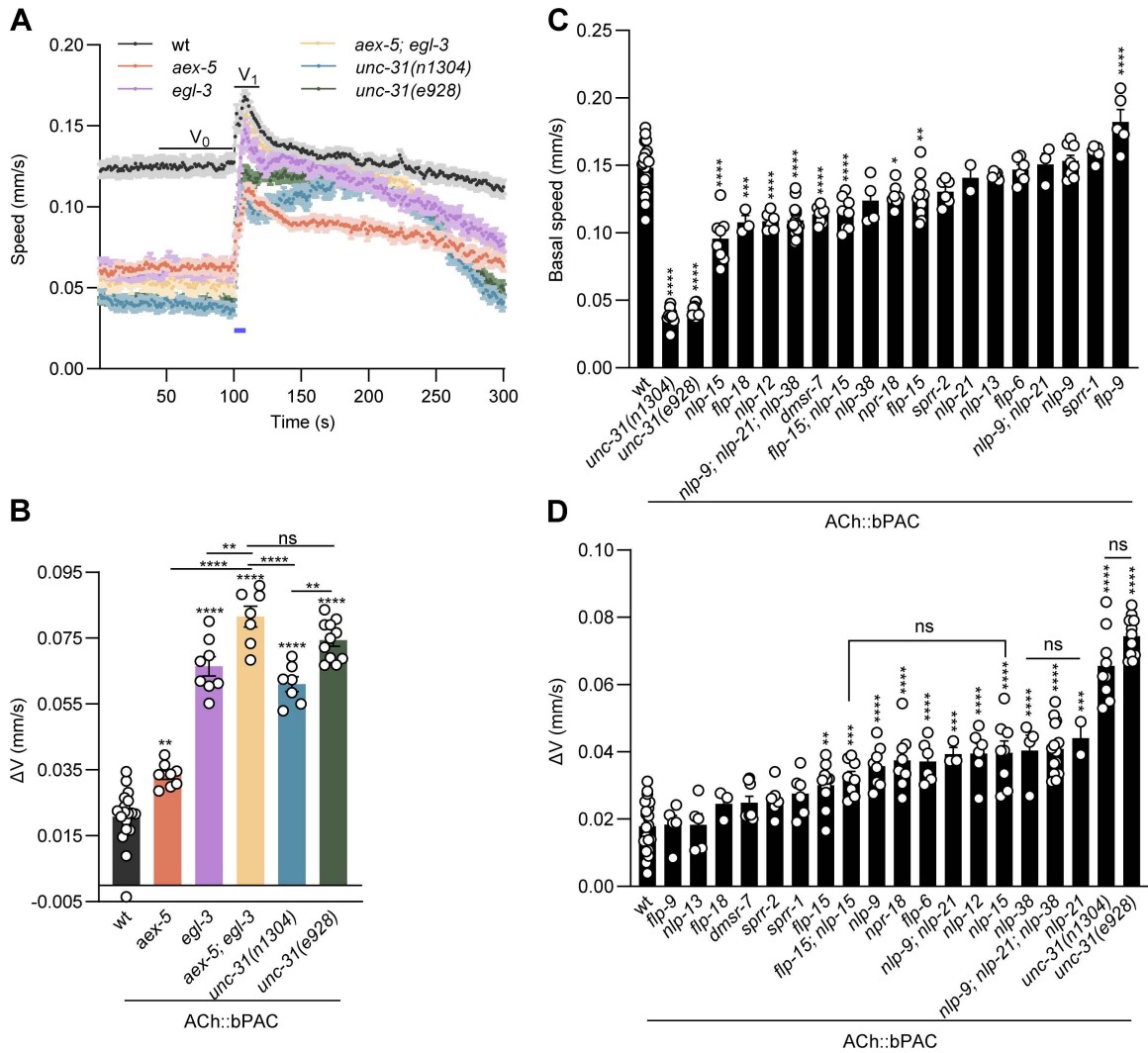

**Fig 3. Candidate screen of neuropeptide/receptor genes affecting cholinergic transmission at the NMJ. (A–D)** Measurements of crawling speed induced by cholinergic bPAC activation. Pro-protein convertase mutants (A, B) and cholinergic-enriched neuropeptides mutants (C, D) were tested. Mean speed traces (A), basal speed comparison of neuropeptides mutants (C), and speed increase comparison ($\Delta V = V_1 - V_0$) after bPAC activation (B, D) in animals of the indicated genotypes (wt, *aex-5, egl-3, aex-5; egl-3, unc-31(n1304), unc-31(e928)* $n = 60-80$, $N = 18, 8, 8, 7, 7, 11$ in (A, B); as well as wt, *nlp-9, nlp-38, sprr-2, sprr-1, flp-6, npr-18, nlp-13, nlp-21, nlp-9; nlp-21, flp-9, flp-18, unc-31(n1304), nlp-12, unc-31(e928), nlp-15, flp-15, flp-15; nlp-15; nlp-9; nlp-21; nlp-38, dmsr-7*, $n = 60-80$, $N = 20, 8, 4, 6, 6, 6, 8, 5, 2, 3, 5, 3, 9, 6, 11, 8, 10, 8, 14, 8$) in (C, D), are shown. For mean speed traces giving rise to group data in C, D, see S3B Fig. Blue bar indicates the 5 s blue light illumination, at 70 μW/mm². All data are presented as mean ± SEM. Statistical significance for multiple-group datasets comparison was determined using one-way ANOVA with Tukey correction. *, **, ***, and **** indicate $p < 0.05$, $p < 0.01$, $p < 0.001$, and $p < 0.0001$, respectively. Numerical data can be found in S3 Data.

## The loss of NLP-9 and NLP-38 in cholinergic neurons affects cholinergic signaling

The absence of a number of neuropeptides abundant in cholinergic neurons had statistically significant effects, including FLP-6, FLP-9, NLP-9, NLP-12, NLP-15, NLP-21, and NLP-38. NLP-12 had been previously studied [22], and FLP-6 is expressed only in VC neurons. NLP-21 is widely used as a marker for DCVs in cholinergic cells [21,48]. Therefore, and because a receptor for NLP-9 is known (NPR-18), we focused on testing the involvement of NLP-9, NLP-21, and NLP-38 neuropeptides in cholinergic transmission in more detail. First, we aimed to demonstrate the sufficiency of these

neuropeptides for overall cholinergic output. The bPAC-induced speed increase that was exaggerated in *nlp-38* and *nlp-9* mutants (ca. 50% of the speed increase in *unc-31* mutants) was rescued by expression from their own promoters (Fig 4A and 4B). Next, we tested muscle activation more directly using ChR2. Using a light intensity of 0.1 mW/mm², *nlp-9* mutants showed a similar relative body length as wt (S4A and S4B Fig). We reasoned that the difference between wt and *nlp-9* mutants may have been masked due to the intense stimulus, as observed in the auxin experiments (S1I and S1J Fig). Thus, we applied a milder stimulation (0.065 mW/mm²), as we observed a significantly enhanced muscle contraction in *snb-1* mutants at this intensity previously [18]. Under these conditions, *nlp-9* mutants showed an enhanced muscle contraction (wt 0.965 relative body length versus 0.93 for *nlp-9*, Fig 4C and 4D). We thus applied this stimulation condition to all the other mutant analyses. *nlp-38* mutants also showed enhanced muscle contraction compared to wt (Fig 4C and 4D), and this was also observed in *nlp-9; nlp-21; nlp-38* triple mutants. However, the triple mutant was not as affected as *unc-31* mutants in the contraction assay (depending on stimulus intensity; S4A and S4B Fig), while it was the strongest mutant along with *nlp-21* in the speed increase assay (Figs 3D and S3B). This indicates complex interplay of the functions of these neuropeptides that are not simply additive.

A cell-type-specific rescue experiment showed that NLP-9 is sufficient for normal evoked muscle contraction in cholinergic, but not GABAergic MNs (Fig 4E and 4F; muscle contraction; S4C and S4D Fig; bPAC-induced speed increase); for NLP-38, we did not observe a cell-type-specific rescue upon testing bPAC induced speed increase (S4E and S4F Fig). We assessed the expression pattern of both neuropeptide precursor genes. While NLP-9 showed a complete overlap with the expression pattern of the vesicular ACh transporter (vAChT) UNC-17 (S5A Fig), *nlp-38* transcription showed a more restricted pattern in only a subset of cholinergic MNs (S5B Fig). NPR-18 has been reported as the receptor for NLP-9 in modulating chemotaxis behaviors [49,51]. Like *nlp-9* mutants, *npr-18* mutants exhibited enhanced evoked muscle contraction (Fig 4G and 4H), and *nlp-9; npr-18* double mutants did not show an additive effect, suggesting that they act in the same pathway. The reported mRNA expression pattern of *npr-18* is inconclusive: scRNAseq data suggests it is expressed in a small subset of cholinergic neurons, VC4 and 5, which innervate vulval muscles, not BWMs. In sum, *nlp-9* is required in cholinergic but not in GABAergic neurons for normal cholinergic-evoked behavior, and is likely detected by NLP-18 receptors.

We also tested the neuropeptides FLP-15 and NLP-15, expressed in GABAergic neurons. Evoked muscle contraction was statistically significantly pronounced in both mutants, though not as much as in *unc-31* animals. *flp-15; nlp-15* double mutants showed a less pronounced effect, though still statistically significantly more than wt animals (Fig 4I and 4J). These non-additive effects indicate functional interplay, that may depend on where the receptors for these neuropeptides are expressed. E.g. NPR-3, expressed in a number of neurons, is one FLP-15 receptor, while a receptor for NLP-15 is as yet unknown [30–32].

### NLP-9 neuropeptides are released from cholinergic neurons

Next, we wanted to assess the release of NLP-9 from cholinergic neurons. To this end, we expressed a mCherry-tagged version of the NLP-9 pro-protein in cholinergic neurons, and imaged the coelomocytes, i.e. cells that take up proteins and other materials from the body fluid (Fig 5A). Fluorescent protein could be observed in these scavenger cells, and the levels of coelomocyte fluorescence were strongly reduced in an *unc-31* mutant background, confirming that the release of NLP-9 from cholinergic neurons is *unc-31* dependent (Fig 5B and 5C). Furthermore, bPAC activation led to a statistically significant increase in the release of cholinergic NLP-9, supporting its role in the modulation of cholinergic signaling (Fig 5B and 5C).

### Neuropeptide signaling affects ACh output at the NMJ

We next analyzed whether neuropeptide signaling by those neuropeptides that showed significant differences in the cAMP stimulation assays (NLP-9, NLP-15, NLP-21, NLP-38, FLP-15), but also for mutations affecting neuropeptide processing

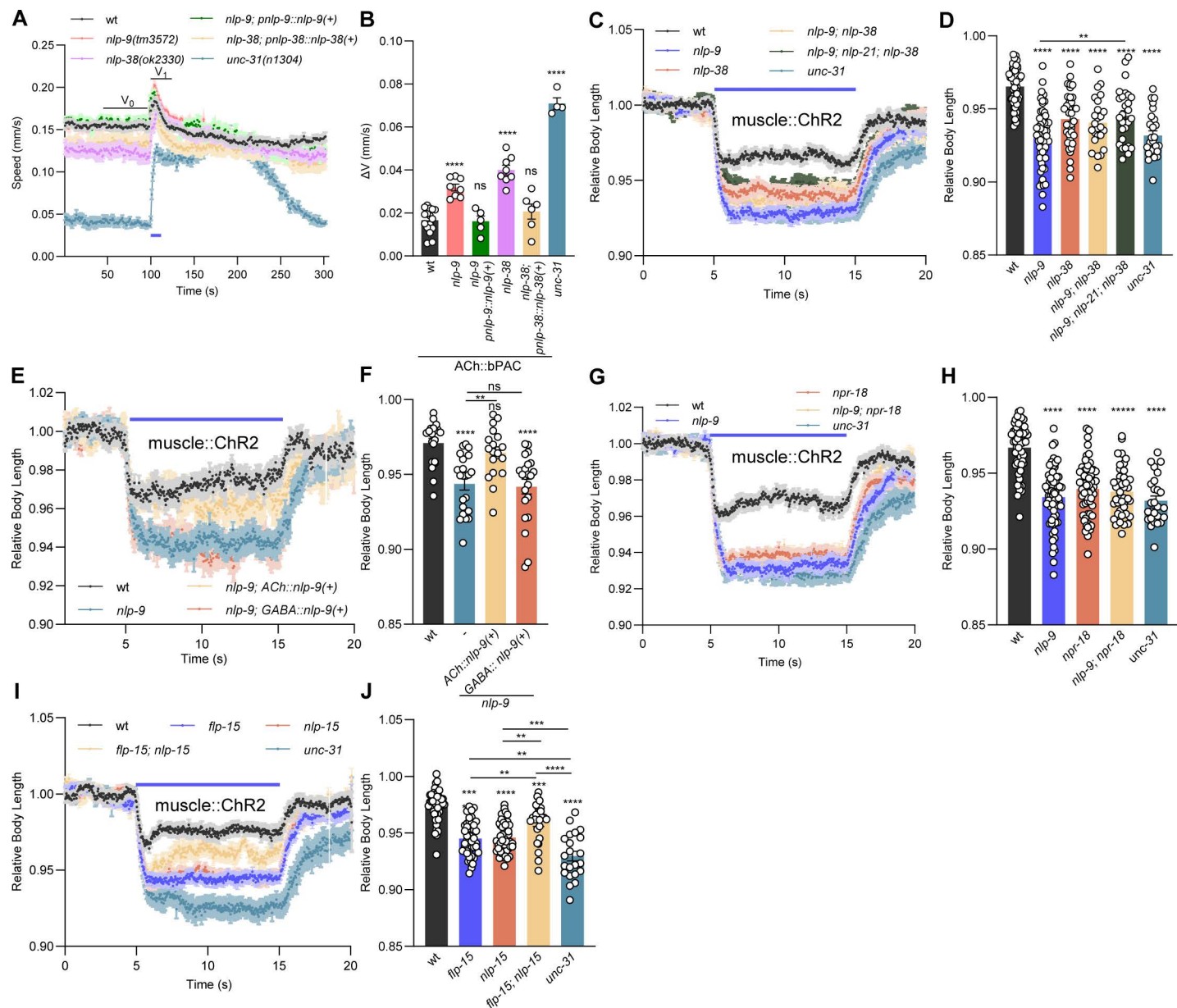

**Fig 4. Loss of NLP-9 in cholinergic neurons induces enhanced evoked muscle activation. (A, B)** Crawling speed induced by cholinergic bPAC activation was compared in the indicated genotypes. Blue bar indicates the 5 s blue light illumination (70 μW/mm²). *nlp-9* and *nlp-38* mutant rescues were achieved by specifically expressing NLP-9 and NLP-38 under their own promoters. Animal number tested in each group: *n* = 60–80, *N* = 16, 8, 5, 8, 6, 4, from left to right, respectively. **(C, D)** Body contraction, induced by muscle::ChR2 activation using 65 μW/mm² blue light stimulation, was compared in the indicated genotypes (wt, *nlp-9, nlp-38, nlp-9; nlp-38, nlp-9; nlp-21; nlp-38, unc-31*, *n* = 53, 57, 38, 25, 27, 26). **(E, F)** As in (C, D), for the indicated genotypes. *nlp-9* mutants were rescued by specifically expressing NLP-9 in cholinergic and GABAergic neurons. Animal number tested in each group: *n* = 20, 22, 18, 22, from left to right, respectively. **(G, H)** As in (C, D), for the indicated genotypes (wt, *nlp-9, npr-18, nlp-9; npr-18, unc-31*, *n* = 57, 63, 60, 41, 26). **(I, J)** Candidate neuropeptides expressed in GABAergic neurons, in addition to cholinergic neurons, were assessed as in (C, D): *flp-15* (*n* = 45) and *nlp-15* (*n* = 47), *flp-15; nlp-15* (*n* = 25), *unc-31* (*n* = 22), compared to wt (*n* = 45). Blue bar indicates the 5–15 s blue light illumination. The data in panels D, F, H, and J represent the mean values over the entire illumination period shown in panels C, E, G, and I, respectively. All data are presented as mean ± SEM. Statistical significance for multiple-group datasets comparison was determined using one-way ANOVA with Tukey correction. **, ***, and **** indicate *p* < 0.01, *p* < 0.001, and *p* < 0.0001, respectively. Numerical data can be found in S4 Data.

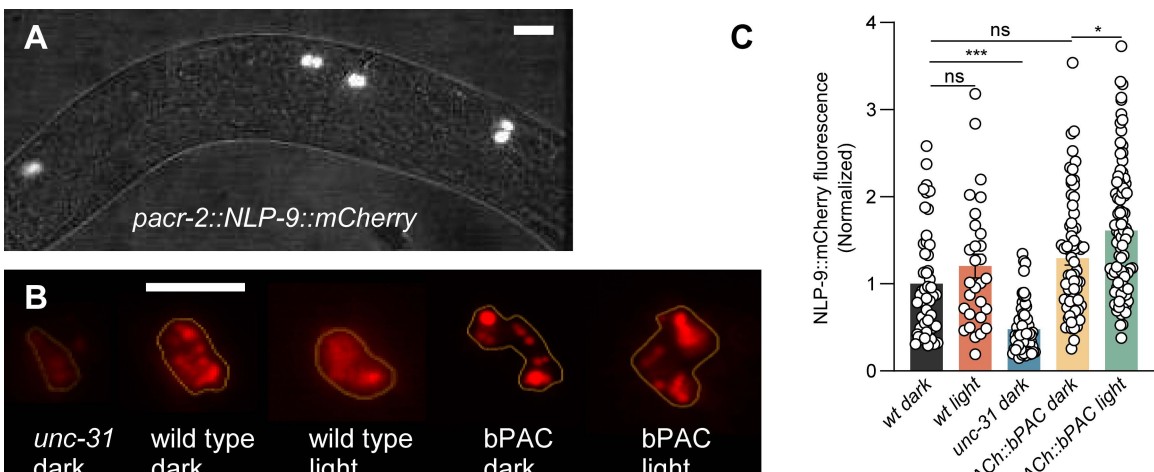

**Fig 5. NLP-9 is released from cholinergic neurons in a cAMP-dependent manner. (A)** Fluorescence of endocytic scavenger cells (coelomocytes) resulting from the secretion of mCherry-tagged NLP-9 pro-protein, expressed in cholinergic MNs (using the *acr-2* promoter), into the body fluid. Scale bar 20 μm. **(B)** Representative images showing NLP-9::mCherry fluorescence in coelomocytes from the indicated genotypes, and with or without blue light stimulation. Scale bar 10 μm. **(C)** Fluorescence quantification of all coelomocytes in each animal, normalized to the signal in wt, is shown for the indicated genotypes and with or without blue light stimulation. Individual data points indicate single coelomocyte cells. Animal number tested *n* = 15, 8, 17, 19, 24, from left to right, respectively. All data are presented as mean ± SEM. Statistical significance comparison was determined using one-way ANOVA with Tukey correction. * and *** indicate *p* < 0.05 and *p* < 0.001, respectively. Numerical data can be found in S5 Data.

(combination of AEX-5 and EGL-3), influences synaptic output. To this end, we recorded postsynaptic currents (PSCs) and analyzed the frequency and amplitude of the recorded events, before, during, and after 5 s bPAC photostimulation of cholinergic MNs. As previously, we observed a robust increase in the frequency of the PSC events in wt animals, since more SVs are primed, and increased amplitude, since the SVs are filled with more ACh; release still depends on intrinsic activity patterns, though [21] (Fig 6A–6C). In *egl-3; aex-5* double mutants, where most of the relevant neuropeptides appear to be missing, there was a noticeable increase in release frequency of SVs (each reported by a single PSC; Fig 6A and 6B). However, this was reduced compared to wt and not statistically significant (Fig 6H). While wt animals exhibited a robust increase in PSC amplitudes within 5–10 s following stimulus onset, to about 150%, we observed only a minor, non-significant increase in *egl-3; aex-5* double mutants (to ca. 110%) (Figs 6A, 6C, 6I, S6A, and S6B). Some of the increase may be due to (unresolved) multivesicular fusion events; we still refer to these as mPSCs, because the underlying SV fusions are not evoked by bPAC stimulation, rather, the induced cAMP signaling affects the rate of priming and mobilization of SVs. Thus intrinsic activity of the neurons leads to more SV fusions per time. In *nlp-9* mutants, amplitudes were reduced though still obviously increased compared to before the light pulse. In *nlp-38* mutants, as well as in *nlp-9; nlp-21; nlp-38* triple mutants, the amplitudes were further reduced and the triple mutant was not obviously exacerbating the phenotype of the single mutant. Overall, the *nlp-9* and *nlp-38* mutants showed statistically significantly different behavior (no amplitude increase and lower frequency increase than wt; Figs 6A, 6D, 6E, 6H, 6I, S6C, and S6D). Thus, NLP-9 and NLP-38 appear to be the main contributors. Together, these findings establish NLP-9 and NLP-38 as regulators of cholinergic signaling at the NMJ, likely through controlling ACh content of SVs [21]. NLP-9 (and likely, NLP-38) is released from cholinergic neurons in a cAMP-dependent manner. The absence of NLP-9 and NLP-38 (along with other neuropeptides) is thus likely to induce compensatory changes that, as in *unc-31* mutants, cause an enhanced behavioral response to (evoked) ACh release.

We also tested animals lacking the neuropeptides expressed in GABAergic (and cholinergic) neurons, i.e. *flp-15; nlp-15* double mutants. These differed from the other mutants in that they showed statistically significantly larger amplitudes

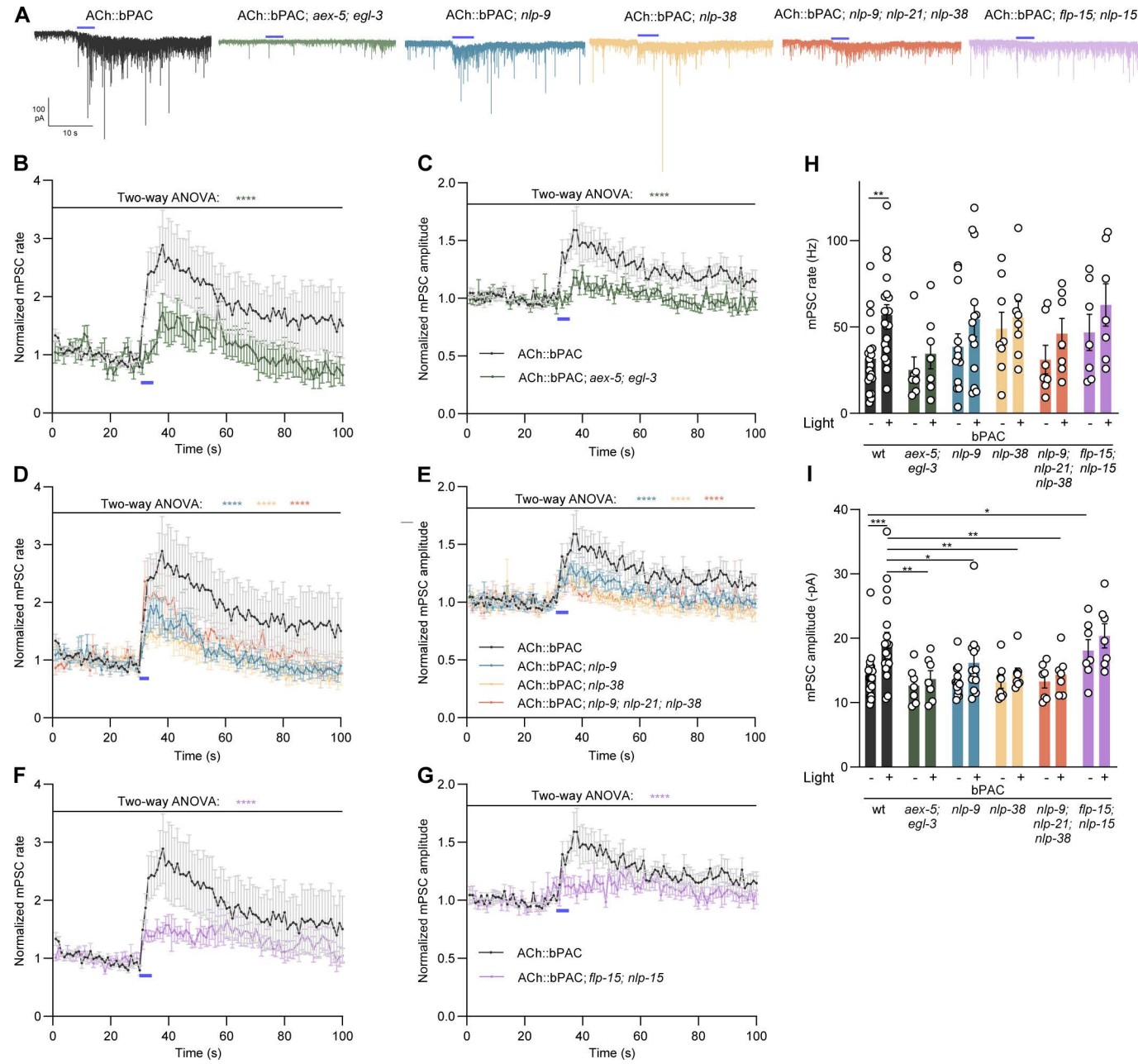

**Fig 6. NLP-9 affects ACh output, along with NLP-38. (A)** mPSCs recorded from the body wall muscle of animals of the indicated genotypes, in response to bPAC photostimulation of cholinergic neurons. Representative traces of mPSCs from the indicated genotypes are shown. Blue bar indicates 5 s blue light stimulation. **(B–G)** Normalized (to the 5 s before light stimulation) mPSC rates (per second; B, D, F) and amplitudes (pA; C, E, G), averaged in 1 s bins. **(H, I)** Group data of the non-normalized data, shown in S6A–S6F Fig. Mean value of before light stimulation (1–30 s) and after stimulation (31–55 s) of mPSC rate (H) and amplitudes (I) are shown. Note, the same wt control data was used in panels B, D, F and C, E, G. Animal number *n* = 19, 7, 13, 8, 7, 7, tested in wt, *aex-5; egl-3, nlp-9, nlp-38, nlp-9; nlp-21; nlp-38,* and *flp-15; nlp-15*, respectively. All data are presented as mean ± SEM. Statistical significance comparison was determined using two-way ANOVA with the Fisher test. *, **, ***, and **** indicate $p < 0.05$, $p < 0.01$, $p < 0.001$, and $p < 0.0001$ respectively. Numerical data can be found in S6 Data.

before the light pulse (Figs 6A, 6F, 6G, 6H, 6I, S6E, and S6F). The amplitudes were only mildly increased by bPAC stimulation, though. These findings indicate that the activity of NLP-15 and/ or FLP-15 may be reducing normal amounts of ACh release, or, that they contribute to a normal balance of up- and downregulation of ACh levels released by cholinergic neurons.

### Postsynaptic compensation in response to reduced cholinergic output

bPAC photostimulation and the induced cAMP signaling causes increased transmitter output, based on mobilization of SVs and their filling with additional ACh [21]. In non-stimulated animals, however, *unc-31* mutants showed reduced miniature PSC amplitudes, though they were not significantly different from wt animals (S7A and S7B Fig), indicating that it requires enhanced neuronal activity to observe changes in ACh output. If the loss of *unc-31* and neuropeptide signaling affects the extent of ACh output, one would expect that also the result of presynaptic depolarization would be affected in *unc-31* mutants. To investigate this, we stimulated cholinergic MNs using ChR2 photoactivation. Previous reports showed that ChR2-evoked [27] and electrical stimulus-evoked ePSCs [26] in *unc-31* mutants are significantly decreased compared to wt. We confirmed this finding, i.e., ChR2 evoked much-reduced ePSCs in muscle (S1C and S1D Fig). Expression of ChR2 itself was not affected by the *unc-31* mutation (S7C and S7D Fig). However, counterintuitively, but consistent with the behavioral results induced by bPAC activation, we observed statistically significantly enhanced muscle contraction in *unc-31* mutants after ChR2 activation, compared to wt (Fig 7A and 7B). Also, muscular Ca$^{2+}$ increase was much larger in *unc-31* mutants (Fig 7C and 7D). Based on these observations, we propose that reduced cholinergic transmission in *unc-31* mutants induces a postsynaptic compensatory mechanism in muscles, leading to enhanced muscle contraction and thus increased effects of evoked ACh release. As a consequence, the smaller presynaptic output in *unc-31* mutants could induce stronger muscle responses, to homeostatically maintain the synaptic strength.

To explore if this hypothesis is correct, we repeated the bPAC stimulation experiment in a reduction-of-function mutant of the choline acetyl transferase, CHA-1. Neurons of this mutant are expected to release much-reduced amounts of ACh [52,53]. The *cha-1(p1152)* animals showed contractions to the same extent as *unc-31* mutants, indicating that reduced ACh output from neurons can induce compensatory changes in muscle that cause increased postsynaptic responses to depolarization (Fig 7E and 7F). To test whether the functionally enhanced muscle contraction and Ca$^{2+}$ activity in *unc-31* mutants results from increased membrane excitability, we applied muscular voltage imaging, using the genetically encoded voltage indicator QuasAr [54,55] (Figs 7G, 7H, and S7C). In response to cholinergic ChR2 stimulation, voltage-dependent fluorescent signals in muscles showed a statistically significantly reduced first peak in *unc-31* mutants, compared to wt (Figs 7G and S7C). This was consistent with the reduced presynaptic ChR2- or electrical-evoked muscle ePSC amplitudes (S1B Fig) [26,27]. However, during the remaining stimulation period (7–10 s), *unc-31* mutants showed increased voltage changes (Fig 7G and 7H), indicating that the BWM plasma membrane became more depolarized during presynaptic stimulation, in line with the idea of enhanced muscle excitability.

BWM depolarization involves the activation of two classes of nicotinic acetylcholine receptors (nAChRs): The heteromeric, levamisole-sensitive L-AChRs, and the homo-pentameric, nicotine-sensitive N-AChRs [56]. Their activation by ACh triggers depolarization and Ca$^{2+}$ entry via the VGCC EGL-19 [57–59]. Possibly, also the T-type Ca$^{2+}$ channel CCA-1 [57] contributes. The enhanced muscle excitability in *unc-31* mutants likely did not occur at the level of nAChRs. As basal mPSC amplitudes were not significantly altered in *unc-31* mutants [21] (S7A and S7B Fig), this suggested that the sensitivity of nAChRs in muscle cells to ACh remained unchanged. Also, when we applied exogenous ACh to BWMs, the evoked (AChR-mediated) postsynaptic currents in wt and *unc-31* animals were indistinguishable (Fig 7I and 7J), indicating that the expression and/or membrane delivery of nAChRs, as well as their functional properties, were unchanged. In sum, the compensatory increase in muscle excitability in *unc-31* animals is not induced at the level of nAChRs.

The enhanced muscle excitability should thus be caused downstream of nAChR activation, in line with the experiments of direct muscle excitation using ChR2. Muscular ChR2 activation evoked statistically significantly increased muscle

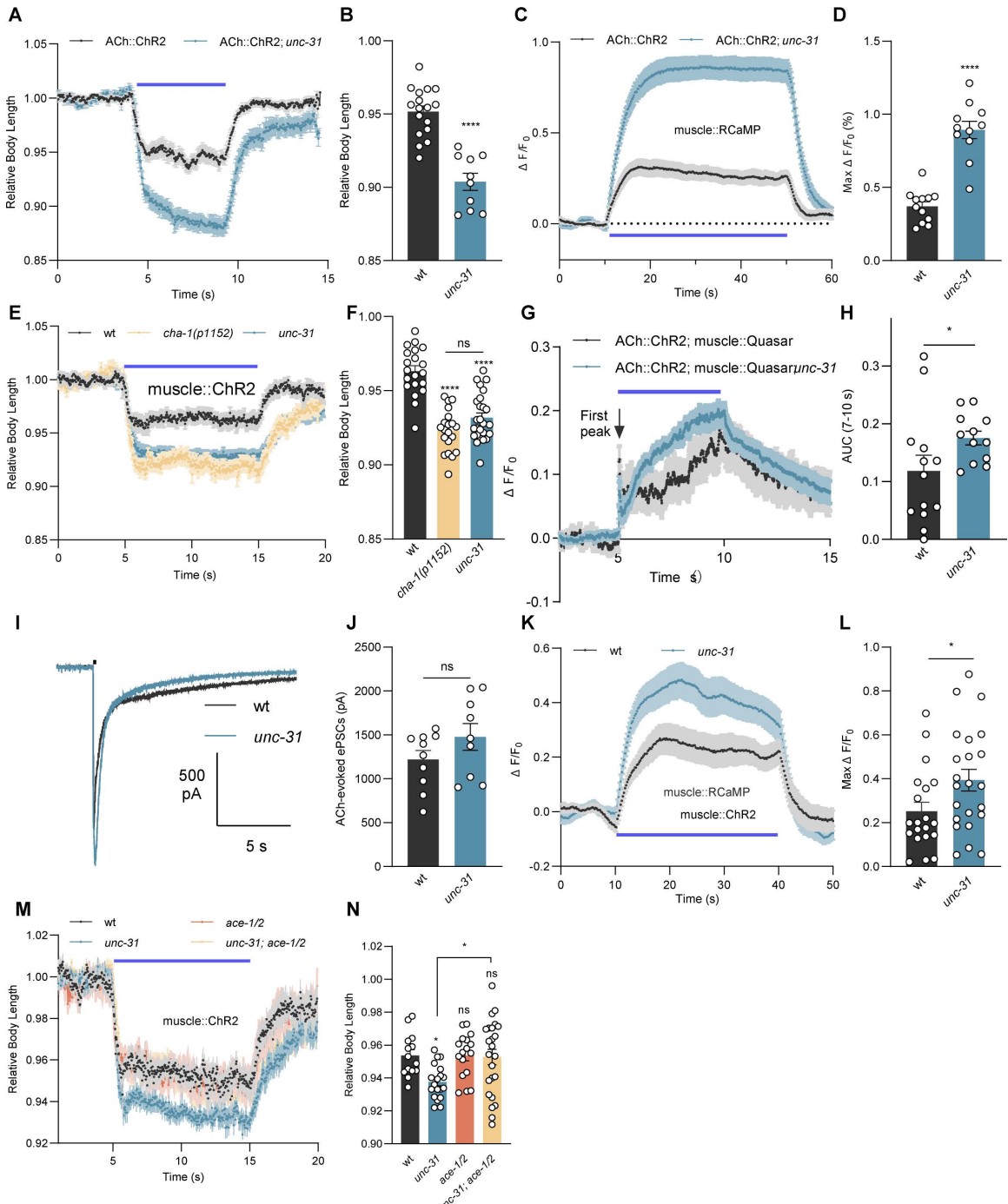

**Fig 7. Reduced cholinergic transmission in *unc-31* mutants induces postsynaptic compensation. (A, B)** Body length change of wt ($n=17$) and *unc-31(n1304)* mutants ($n=10$), induced by ChR2 activation in cholinergic neurons. **(C, D)** RCaMP calcium imaging in BWM cells of wt ($n=12$) and *unc-31(n1304)* mutants ($n=11$), signal increase evoked by ChR2 stimulation of cholinergic neurons. Max $\Delta F/F_0$ indicates the maximal signal observed after ChR2 stimulation. **(E, F)** Body length change of wt ($n=20$), *cha-1(p1152)* ($n=20$), and *unc-31(n1304)* mutants ($n=26$), induced by ChR2 activation in muscles. **(G, H)** Voltage imaging using the fluorescent voltage indicator QuasAr, expressed in BWM cells of wt ($n=13$) and *unc-31* mutants ($n=12$), before, during, and after depolarization evoked by ChR2 stimulation of cholinergic neurons. QuasAr fluorescent signal is normalized as $\Delta F/F_0$, where $F_0$ is the average signal during the first 5 s before blue light illumination, and $\Delta F$ is the difference between the fluorescent signal $F$ and $F_0$. AUC (7–10 s) indicates the area under the curve during the stimulation period following the first peak (H). For analysis of the first peak, see S7E Fig. **(I, J)** ACh-evoked currents recorded from BWM cells of wt ($n=10$) and *unc-31* mutants ($n=9$). Representative traces (I) and group data of peak currents (J) are shown. **(K,**

**L)** RCaMP calcium imaging in BWM cells of wt ($n = 20$) and *unc-31* mutants ($n = 23$), signals evoked by muscular ChR2 stimulation. Max $\Delta F/F_0$, maximal signal after ChR2 stimulation. **(M, N)** Body contraction induced by muscular ChR2 activation was compared in the indicated genotypes. $n = 14, 19, 17, 24$ animals tested of wt, *unc-31, ace-1; ace-2*, and *unc-31; ace-1; ace-2* animals, respectively. In all experiments except (I, J), ChR2 (in cholinergic neurons or muscles) was activated using 65 µW/mm² blue light illumination. The data in panels B, F, and N represent the mean values over the entire illumination period shown in panels A, E, and M, respectively. All data are presented as mean ± SEM. Statistical significance for two-group datasets and multiple-group datasets comparison was determined using unpaired *t* test and one-way ANOVA with Tukey correction, respectively. *, **, and **** indicate $p < 0.05$, $p < 0.01$, and $p < 0.0001$, respectively. Numerical data can be found in S7 Data.

contraction (Fig 1G and 1H) and Ca²⁺ increase (Fig 7K and 7L) in *unc-31* mutants, compared to wt. Thus, muscle Ca²⁺ influx, as a proxy for excitability, appears to be enhanced in *unc-31* mutants, possibly in response to the reduced cholinergic transmission. To further validate the causality between the reduced cholinergic transmission and enhanced muscle excitability, we genetically manipulated presynaptic ACh levels in *unc-31* mutants. AChEs encoded by *ace-1* and *ace-2* break down ACh in the synaptic cleft. Consequently, AChE mutants should cause ACh accumulation in cholinergic synapses, and previous analyses demonstrated locomotion defects [60,61]. Introducing *ace-1; ace-2* double mutations in the *unc-31* background indeed reverted the increased muscle excitability to wt level as the *unc-31; ace-1; ace-2* triple mutants showed a similar extent of muscle contraction as wt (Fig 7M and 7N). However, *ace-1; ace-2* double mutants showed no significantly different muscle contraction compared to wt, suggesting that an elevated presynaptic ACh level, contrary to its reduction, does not alter evoked muscle contractions. This is in contrast to an earlier report describing upregulation of ACh output in response to pharmacological AChE blockade [22]. However, these researchers used an acute assay, while in our experiment, mutants were used that may have undergone adaptation. Collectively, our results implicate that, in response to the decreased presynaptic ACh output in *unc-31* mutants, muscle excitability is homeostatically increased.

### Postsynaptic compensation involves EGL-19/CaV1 L-type VGCCs

How is the muscle homeostasis responding to the loss of neuropeptidergic regulation achieved? As we had excluded the involvement of nAChRs in the homeostatic mechanisms (Fig 7I and 7J), we next assessed EGL-19 L-type VGCCs. EGL-19 functions in the presynaptic terminal to regulate spontaneous release [24,62]. In muscles, it regulates Ca²⁺-dependent APs and triggering of the ryanodine receptor UNC-68, as well as the muscle contractile apparatus [57,58]. Since *egl-19* null mutants are inviable, we acutely applied the specific antagonist nemadipine to inhibit the function of *egl-19* [63]. Nemadipine treatment statistically significantly decreased the muscle contraction of both wt and *unc-31* mutants, elicited by muscle ChR2 activation (Fig 8A and 8B). More importantly, after nemadipine treatment, *unc-31* mutants showed muscle contraction that was not significantly different from that of wt. We further confirmed the role of *egl-19* in regulating muscle excitability by (partial) knockdown of *egl-19* via RNA interference (RNAi) [64,65]. Specific *egl-19* RNAi abolished the difference in muscle contraction between wt and *unc-31* mutants (Fig 8C and 8D). Together, these results suggest that *egl-19* may be required for the enhanced muscle excitability in *unc-31* mutants, though this interpretation is complicated because EGL-19 is also a main mediator of muscle contraction. Nemadipine treatment also diminished the difference of muscle contraction between wt and *aex-5* mutants, implicating the involvement of AEX-5 in regulating muscle excitability (S8A Fig and Fig 8B). Last, when we stimulated the presynaptic cholinergic neurons by ChR2, the enhanced body contraction seen in *unc-31* mutants was decreased to wt level by *egl-19* RNAi (Fig 8E and 8F). In sum, our results suggest that the enhanced muscle excitability induced by the loss of neuropeptidergic regulation is mediated by EGL-19.

### EGL-19 expression is upregulated in *unc-31* mutants and promotes postsynaptic excitability

To investigate how the EGL-19 channel may be regulated in response to the loss of neuropeptides, and as a means to achieve homeostatic compensation, we first looked at its expression levels in wt and *unc-31* mutants. Endogenous EGL-19 in muscle cells was labeled using the split mCherry complementary system (a gift from Joshua M. Kaplan) [66–68]. We

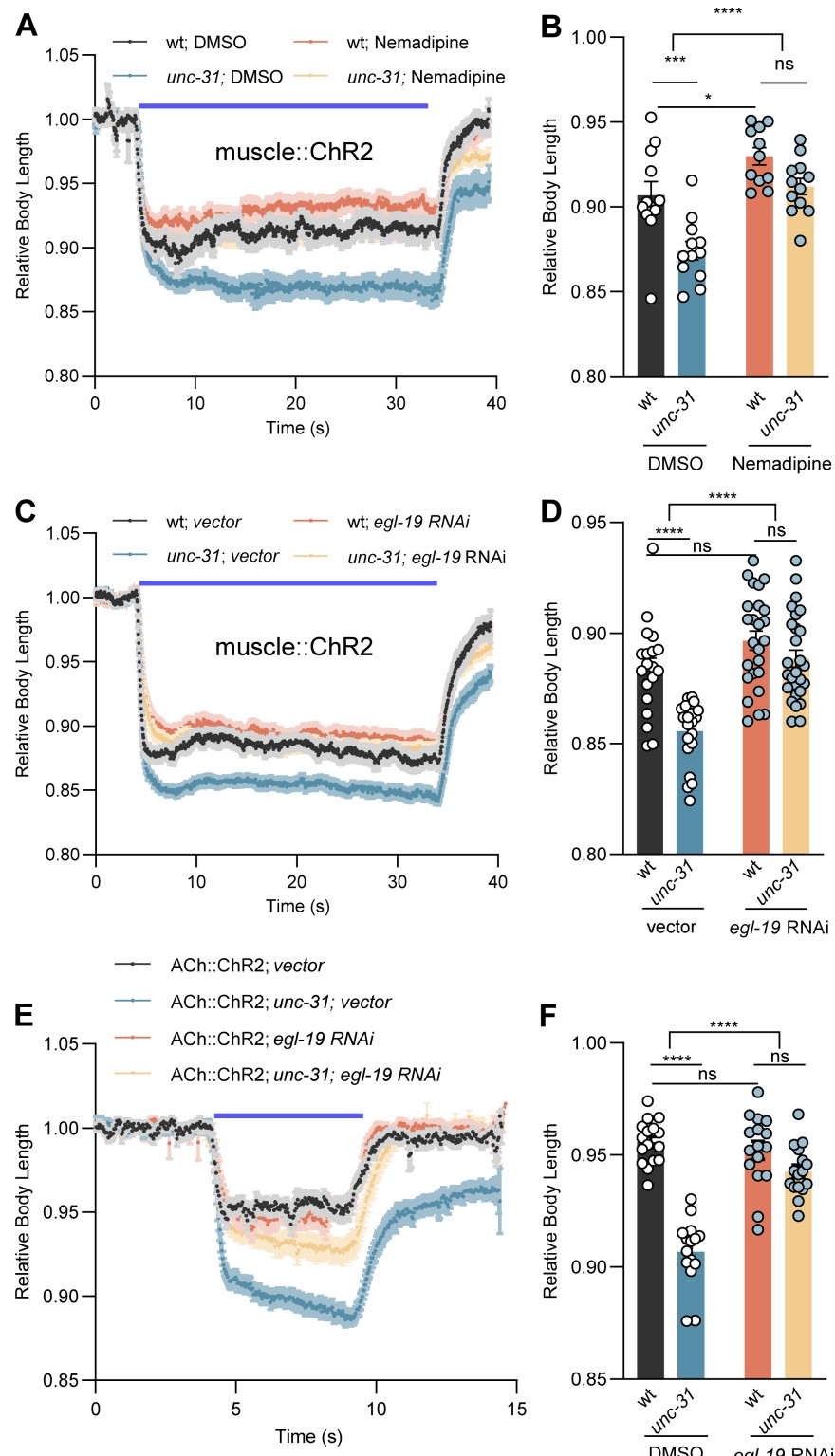

**Fig 8. EGL-19 CaV1 mediates postsynaptic compensation of reduced presynaptic ACh release. (A–D)** Body contraction induced by muscular ChR2 activation using 1.1 mW/mm² blue light stimulation was compared in the indicated genotypes and treatments. Relative body lengths after application of the EGL-19 CaV1 specific inhibitor nemadipine (A, B) and specific *egl-19* feeding RNAi (C, D) treatment are shown. DMSO-treated animals

and animals fed the empty vector L4440 were used as controls. Note that in the early stage of our study, we applied a 30 s (5–35 s) stimulation using a saturating light intensity of 1.4 mW/mm². However, since worms reached maximum muscle contraction already after 1 s stimulation, we shortened the stimulation time to 10 s (5–15 s), allowing us to optimize experimental efficiency. Also, we reduced the stimulation intensity to 0.065 mW/mm² to better reveal the smaller differences of milder mutants. **(E, F)** Body contraction induced by ChR2 activation in cholinergic neurons using 0.065 mW/mm² blue light was compared in the indicated genotypes or RNAi treatments. Relative body lengths after specific *egl-19* RNAi treatment are shown. Number of animals tested in (B) $n = 12, 12, 11, 12$; (D) $n = 19, 21, 25, 27$; (F) $n = 16, 15, 16, 16$, from left to right, respectively. The data in panels B, D, and F represent the mean values over the entire illumination period (5–35 s) shown in panels A, C, and E, respectively. All data are presented as mean ± SEM. Statistical significance was determined using one-way ANOVA with Tukey correction and two-way ANOVA with Sidak's multiple comparisons test. *, ***, and **** indicate $p < 0.05$, $p < 0.001$, and $p < 0.0001$, respectively. Numerical data can be found in S8 Data.

observed a statistically significant increase in the mCherry fluorescence in *unc-31* mutants, indicating that the expression level of EGL-19 is upregulated (Fig 9A and 9B). We performed two additional experiments: First, we asked if the lack of the cholinergic neuropeptide NLP-9 is sufficient to affect muscular EGL-19 expression levels. However, this was not the case (S9A and S9B Fig), arguing that it requires the absence of multiple neuropeptides to induce the postsynaptic alteration. Second, we asked if the effect of *unc-31* on muscular EGL-19 levels is cell-autonomous, or rather ubiquitously affecting *egl-19* gene expression. Thus, we quantified expression of EGL-19 in cholinergic and GABAergic MNs. Interestingly, we observed a statistically significant upregulation of EGL-19 levels in cholinergic neurons (S9C and S9D Fig). In contrast, we did not observe any change in GABAergic neurons. These findings imply that the *unc-31* phenotype at the NMJ may comprise a presynaptic and a postsynaptic component. Reduced levels of EGL-19 in cholinergic neurons were previously shown to affect the rate of cholinergic SV fusions, but not the amplitude of cholinergic mPSCs [62]. Thus, upregulation of EGL-19 in cholinergic neurons may be expected to increase mPSC rates. However, we previously found reduced SV fusion rates in *unc-31* mutants [21]. In addition, the postsynaptic increase in excitability may be due to EGL-19 upregulation.

To further confirm that EGL-19 regulates evoked muscle contractions, we overexpressed it in muscles and measured the body length change evoked by ChR2 activation in muscle. Animals with EGL-19 overexpression displayed statistically significantly stronger body contraction during ChR2 activation compared to wt, confirming that an elevated EGL-19 level induces higher muscle excitability (Fig 9C and 9D). EGL-19 regulates APs in muscles [57–59]. If EGL-19 expression was upregulated in *unc-31* mutants, we would expect the APs to be potentiated as well. Consistent with this idea, in response to injecting current steps (20 pA, 50 ms) in patch-clamped muscle cells, the amplitude of induced APs was higher in *unc-31* mutants compared to wt (Figs 9E, 9F and S9G–S9I). However, in animals overexpressing EGL-19, this was not the case. Yet, EGL-19 overexpression caused the muscles to respond differently to the current step: The voltage increases started much delayed (Figs 9E, 9G and S9G–S9I) and the APs were longer lasting (Figs 9E, 9H and S9G–S9I). This led to a larger area under the curve for APs in EGL-19 overexpression animals (Figs 9E, 9I and S9G–S9I). Taken together, these results indicate that *egl-19* may play an important role in mediating postsynaptic compensation, which is triggered by the loss of presynaptic neuropeptidergic modulation.

## Discussion

Neuropeptidergic modulation of chemical synaptic transmission is likely to be a widely occurring phenomenon, and given the large variety of neuropeptide/neuropeptide-receptor systems, many mechanisms may exist. Here, we have analyzed details of how the NMJ of the nematode *C. elegans* uses neuropeptides to regulate the output of ACh, and possibly GABA, to fine-tune transmission, but likely also to generally set the transmission output level. In the absence of neuropeptides, or when specific neuropeptide species, identified in this work, are missing, less transmitter output occurs, which leads to an upregulation of postsynaptic contractions, as a proxy for muscle excitability. We showed that this likely occurs through the increased expression of the CaV1 VGCC EGL-19. This way, the lack of neuropeptides, by effecting reduced ACh signaling [21], causes the muscle to produce larger APs, to balance the missing cholinergic input. How the

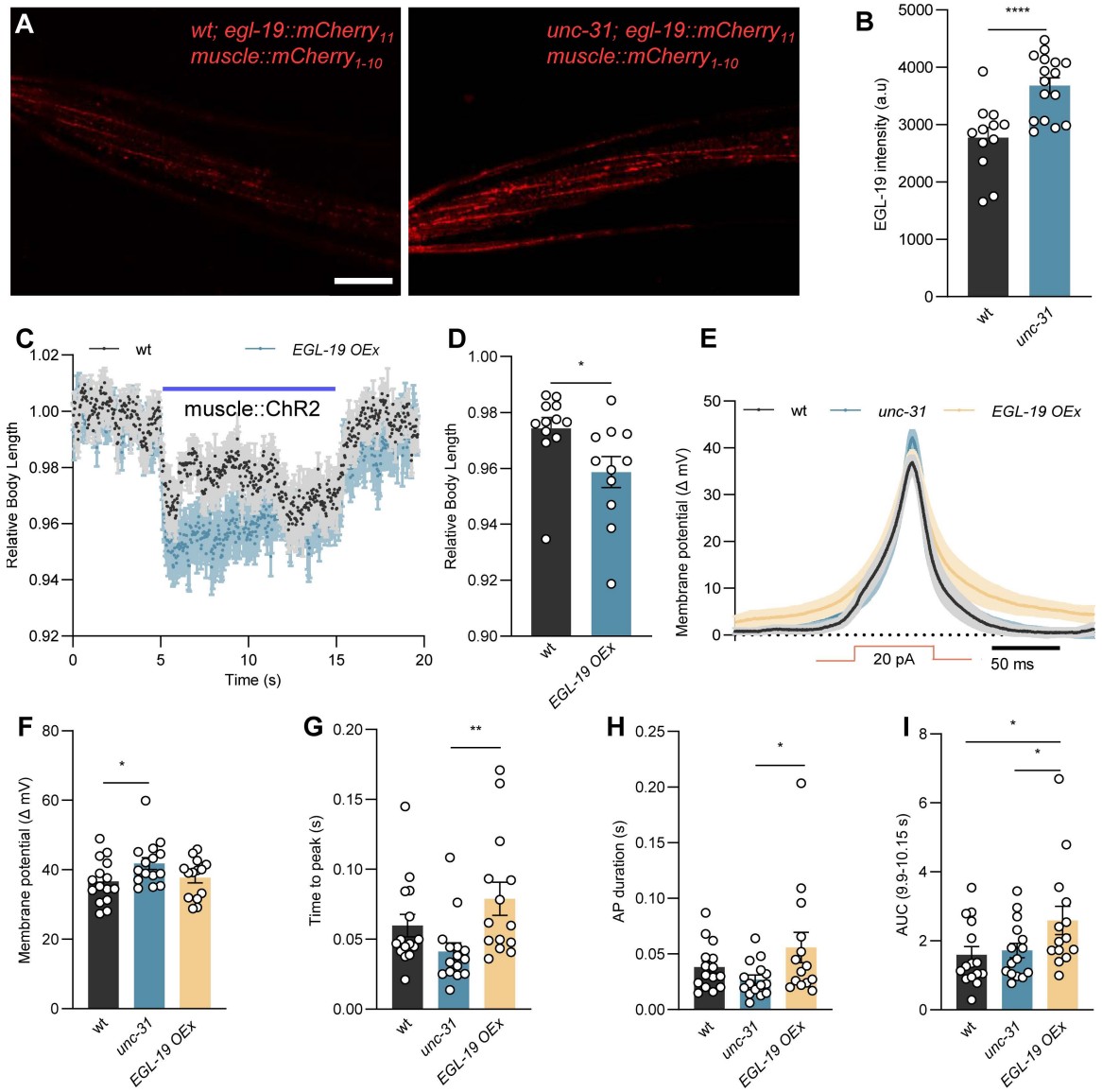

**Fig 9. EGL-19 CaV1 is upregulated in *unc-31* mutants, leading to enhanced postsynaptic excitability. (A, B)** Endogenous expression of EGL-19 in muscle cells was quantified in wt (*n* = 12) and *unc-31* mutants (*n* = 16). Representative images (A; Scale bar 20 µm) and summary data (B) are shown. **(C, D)** Body contraction induced by muscular ChR2 activation using 65 µW/mm² blue light stimulation. wt (*n* = 12) and muscle-specific EGL-19 overexpression (OEx) animals (*n* = 11) were compared. The data in panel D represent the mean values over the entire illumination period (5–15 s) shown in panel C. **(E–I)** Current step-induced BWM membrane potential changes were compared in the indicated genotypes. Mean ± SEM membrane potential changes (E) and summary data (F), mean time from current injection onset to reaching peak potentials (G), duration of the induced action potentials (H), and area under the curve (of the 250 ms time window, centered on the peak) (I) are shown. Numbers of animals tested in (F–I) *n* = 15, 15, 14, from left to right, respectively. All data are presented as mean ± SEM. Statistical significance for two-group datasets and multiple-group datasets comparison was determined using unpaired *t* test, one-way ANOVA with uncorrected Fisher's LSD test, or Kruskal–Wallis test with Dunn's comparison, respectively. *, **, and **** indicate $p < 0.05$, $p < 0.01$, and $p < 0.0001$, respectively. Numerical data can be found in S9 Data.

upregulation of CaV1 is achieved is currently unclear. It may involve previously discovered pathways for muscular regulation of excitability [23–25], though these involved the expression levels of nAChRs, which we did not find to be upregulated. Yet, gene expression mediated by the MEF-2 transcription factor might also affect EGL-19 expression. We propose

that the neuropeptides work in an autocrine fashion to regulate presynaptic ACh output, however, it is possible that they also have postsynaptic effects. The latter appears unlikely, though, unless these peptides act inhibitory on (*egl-19*) gene expression. Likewise, we do not know how autocrine neuropeptide signaling in cholinergic neurons affects the function of the vAChT, or how paracrine signaling of neuropeptides from GABAergic neurons to cholinergic neurons may reduce their activity. However, it is probably not via reducing cAMP levels (G$\alpha_{i/o}$ signaling), as in our bPAC stimulation experiments, cAMP levels are strongly increased in cholinergic neurons.

Both cholinergic and GABAergic MNs are involved in neuropeptidergic regulation of the NMJ, as a complete rescue of *unc-31* deletion was achieved only if the CAPS protein was expressed in both cell types. Furthermore, the PC AEX-5 rescue was only observed in GABAergic neurons. Thus, there is likely a complex interplay of neuropeptidergic regulation at the NMJ [32]. Interestingly, the double mutants *flp-15; nlp-15*, affecting neuropeptides expressed in GABAergic neurons in addition to cholinergic neurons, exhibit statistically significantly increased postsynaptic mPSCs, indicating that they mediate inhibitory effects, most likely on cholinergic MNs. The neuropeptidergic interplay of the MN classes is indicated by the large number of neuropeptides as well as neuropeptide receptors that are expressed in the different MN types [29,30]. Deletion of several of those neuropeptides (NLP-9, -12, -15, -21, -38, FLP-6, -9, -15, -18), and of the NPR-18 receptor, affected (optogenetically induced) locomotion or muscle contraction. For NLP-9 and NLP-38, we could show by electrophysiology that there is an abolishment of the cAMP-induced increase of mPSC amplitudes. Several of these neuropeptides may be part of autocrine feedback loops (NLP-9, -12, FLP-6, -9, -15, -18), as they are expressed along with their cognate receptors in MNs [31,32]. However, often these peptides, or some of the peptides that are part of the individual precursor proteins, bind to a number of different receptors, expressed also in other parts of the nervous system, complicating interpretation of the findings. We tested the effects of mutations in NPR-18 (putative receptor for NLP-9), SPRR-1, and SPRR-2 (receptor for NLP-38). Only NPR-18 had effects that were consistent with NLP-9 effects. However, the expression of NPR-18 mRNA [30] is found in only the VC4 and VC5 neurons that do not innervate body wall muscles. SPRR-2 is also not obviously expressed in cholinergic neurons. The receptors for NLP-12, CKR-1, and CKR-2 [22,31,69], however, are expressed in almost all classes of cholinergic MNs. As we have shown previously, neuropeptides released from cholinergic neurons (this may include DVA, releasing NLP-12), affect the filling state of cholinergic SVs [21]. Here, we identified these neuropeptides, thus one can now address how activation of neuropeptide receptors (and which ones) regulates filling of SVs via the vAChT UNC-17. Furthermore, we identified the requirement of PCs AEX-5 and EGL-3 for processing of the neuropeptides. Assessment of the expected cleavage sites in the neuropeptides we studied predicts a need for AEX-5 only in the case of NLP-12, while all others require EGL-3 (S1 Table). Yet, NLP-12 is not predicted to be expressed by GABAergic neurons. Thus, more work is needed to clarify these details. However, the cleavage specificity of individual neuropeptide precursors and the need for a specific convertase should be confirmed by proteomics analyses of the respective mutants [39].

The postsynaptic compensation we observed in muscle in the absence of neuropeptides appears to occur via upregulation of the CaV1 VGCC EGL-19. In line with this, *unc-31* mutants showed increased AP amplitudes. We explored if this could be mimicked by overexpressing EGL-19. While we observed increased muscle contraction, just as in *unc-31* mutants, we did not observe an increased AP amplitude. However, APs were longer lasting, with a delayed onset and a larger AP integral, demonstrating that expression levels of EGL-19 can affect muscle excitability. Yet, neuropeptidergic regulation (or, the absence thereof) may induce additional modes of regulation of the CaV1 channel, not mimicked by its overexpression. In addition, also other channels may contribute, like voltage-gated potassium channels that are required for the AP down stroke (SHK-1, SLO-2; [57,58]), and the ER-resident ryanodine receptor, encoded by *unc-68*, may effect increased Ca$^{2+}$ levels. The comparably mild effects of *egl-19* RNAi or nemadipine treatment likely are due to the fact that EGL-19 is essential and thus animals with a full knockdown would die, and the individuals assayed are likely showing only a mild depletion. Previous work used nemadipine in dissected animals [62], showing an almost complete abolishment of currents and synaptic activity. However, we likely cannot reach comparable concentrations in intact animals, or would have to increase the amounts such that the DMSO vehicle itself becomes toxic.

In sum, our data are in agreement with the following ideas: Neuropeptides released from cholinergic and GABAergic neurons, at least partially, act in an autocrine fashion in cholinergic neurons, but there is also likely a feedback loop from GABAergic to cholinergic neurons, to affect the output of ACh (Fig 10). The lack of this signaling appears to cause reduced SV filling with ACh [21] and thus reduced ACh output in *unc-31* mutants. The reduced ACh output and thus reduced muscle depolarization seem to cause a compensatory upregulation of EGL-19 CaV1 expression, leading to more $Ca^{2+}$ entry despite less initial depolarizing signal from nAChRs. Whether such mechanisms are conserved in other animals will have to be demonstrated. Recently, we studied the larval zebrafish NMJ, in which optogenetically induced cAMP increase via bPAC also caused activation of locomotion behavior [70]. Neuropeptide-processing mutants lacking carboxypeptidase E, interestingly, also showed an increased response to the optogenetic stimulus. This, however, seemed to be affected on the level of nAChR expression in muscle cells, unlike in *C. elegans*, while the role of CaV1 channels has not been analyzed as yet.

## Materials and methods

### Worms and maintenance

*C. elegans* strains were cultivated at 20 °C, on nematode growth media (NGM) plates seeded with *Escherichia coli* strain OP50-1 as previously described [71]. The following strains were generated or used:
**EG5027**: *oxIs353[myo-3p::ChR2::mCherry; unc-54UTR; lin-15(+)] V*, **EN8636:** *kr462[daf-2::linker::AID::linker:: mNeonGreen] III; krSi234[Punc-17::TIR1::BFP::unc-54 3′ UTR]*, **KP9876**: *egl-19(nu674[EGL-19 GFP11 knock-in]);*

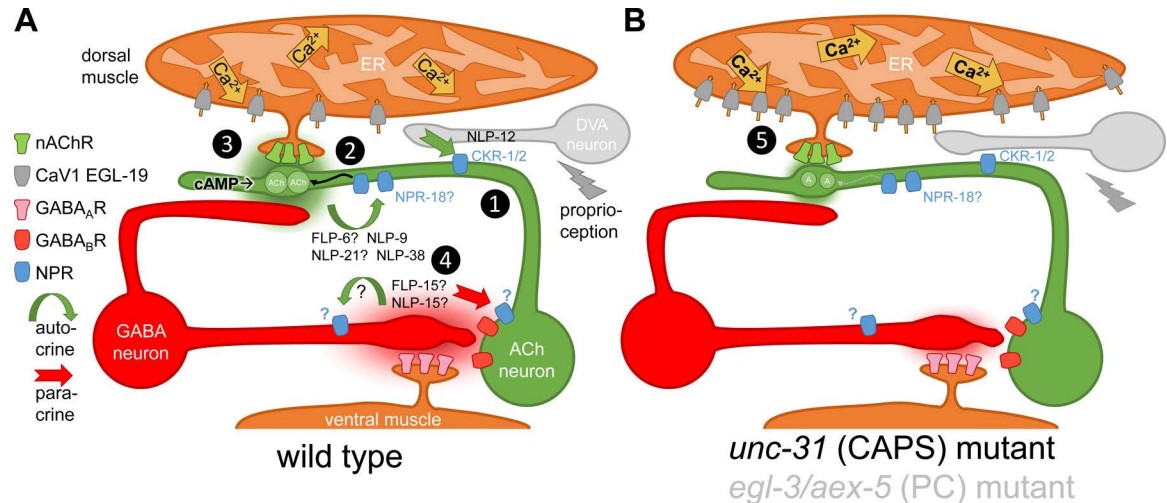

**Fig 10. Autocrine and paracrine neuropeptidergic signaling at the *Caenorhabditis elegans* NMJ affects cholinergic transmission.** Neuromuscular junction of *C. elegans*, with cholinergic and GABAergic motor neuron, as labeled, innervating dorsal and ventral muscle, respectively. Dyadic synapse of cholinergic neuron to GABAergic neuron and muscle. Note that the same innervation pattern exists from different sets of MNs, but with inverted topology (not represented in this graph). Also shown is the proprioceptive neuron DVA, that innervates cholinergic neurons along the ventral nerve cord (drawn more dorsal, for simplicity). Relevant ion channels and receptors are indicated on the left (NPR: neuropeptide receptor). In wt **(A)**, DVA releases NLP-12 neuropeptides onto cholinergic neurons, detected by CKR-1 and -2 receptors (1). (2) Cholinergic neuron releases ACh (green cloud) and neuropeptides FLP-6, NLP-9, NLP-21, and NLP-38, (likely) in an autocrine fashion, on NPR-18 and other, unknown neuropeptide receptors on cholinergic neuron. Activation of these receptors induces additional ACh filling of SVs (curved black arrow). (3) ACh release on muscle causes depolarization and $Ca^{2+}$ influx into the cytosol, through CaV1 channels and from the ER (through UNC-68 ryanodine receptors, not represented in this graph); ACh release also activates GABAergic neuron. (4) GABA and neuropeptides FLP-15 and NLP-15 are released on muscle and cholinergic neurons, and are detected by GABA$_A$ (muscle) and GABA$_B$ receptors (cholinergic neuron), as well as by unknown neuropeptide receptors; signaling to ACh neurons is paracrine. In *unc-31* (CAPS) or *egl-3; aex-5* (pro-protein convertases, PCs) double mutants **(B)**, no mature neuropeptides are released or formed, thus the signaling steps induced by neuropeptides are absent. As a consequence, (5) less ACh is filled into SVs and released from cholinergic neuron. In response, compensatory upregulation ensures homeostatic NMJ signaling by providing more $Ca^{2+}$ influx and thus depolarization of muscle cell.

*nuSi250[punc-129::GFP1−10::SL2::UNC-57::mCherry]*, **KP11046:** *egl-19(nu674[EGL-19 GFP11 knock-in]); nuSi285[punc-47::GFP1-10::SL2::UNC-57::mCherry]*, **KP10154:** *egl-19(nu698[EGL-19 mCherry11 knock-in]) IV; nuTi335[ppat-10::sfCherry1-10]*, **RJP5269:** *rp166(unc-31-linker-GFP-TEV- AID-FLAG) IV*, **ZX460:** *zxIs6[punc-17::chop-2(H134R); lin-15+]*, **ZX1460:** *zxIs53[punc-17::bPAC::YFP; pmyo-2::mCherry]*, **ZX1659:** *zxIs52[pmyo-3::RCaMP35]; zxIs6*, **ZX1869:** *zxIs53; zxIs6; zxIs52*, **ZX1870:** *unc-31(n1304) IV; zxIs53*, **ZX2482:** *zxEx1139[pmyo-3::QuasAr; pmyo-2::CFP]; zxIs5[punc-17::chop-2(H134R); lin-15+]*, **ZX2640:** *nlp-21(tm2569) III; zxIs53*, **ZX2684:** *flp-18(gk3063) X; zxIs53*, **ZX2719:** *unc-31(n1304) IV; zxIs6; zxIs52*, **ZX2720:** *unc-31(n1304) IV; zxIs53; zxIs6; zxIs52*, **ZX2771:** *unc-31(n1304) IV; oxIs353*, **ZX2772:** *unc-31(n1304) IV; zxEx1139[pmyo-3::QuasAr; pmyo-2::CFP]; zxIs5[punc-17::chop-2(H134R); lin-15+]*, **ZX2851:** *flp-9(ok2730) IV; zxIs53*, **ZX2852:** *nlp-38(ok2330) I; zxIs53*, **ZX2870:** *flp-6(ok3056) V; zxIs53*, **ZX2877:** *npr-18(ok1388) X; zxIs53*, **ZX2879:** *sprr-1(tm3658) X; zxIs53*,  **ZX2881**: *nlp-9(tm3572) V; zxIs53*, **ZX2932:** *nlp-38(ok2330) I; zxEx1538[pnlp-38::nlp-38::SL2::mCherry, pmyo-2::CFP]; zxIs53*, **ZX2933:** *sprr-2(ok3290) X; zxIs53*, **ZX2981:** *nlp-13(sy1453) V; zxIs53*, **ZX2982:** *nlp-9(tm3572) V; zxEx1535[pnlp-9::nlp-9::SL2::mCherry, pmyo-2::CFP]; zxIs53*, **ZX3039:** *zxEx1537[pacr-2::nlp-9::mCherry; pmyo-2::CFP]; zxIs53*, **ZX3045:** *nlp-9(tm3572) V; oxIs353*, **ZX3099:** *nlp-9(tm3572) V; nlp-21(tm2569) III; zxIs53*, **ZX3119:** *zxEx1537[pacr-2::nlp-9::mCherry; pmyo-2::CFP]*, **ZX3248:** *nlp-9(tm3572) V; nlp-21(tm2569) III; nlp-38(ok2330) I; oxIs353*, **ZX3258:** *unc-31(n1304) IV; zxIs6*, **ZX3260:** *nlp-38(ok2330) I; oxIs353*, **ZX3303:** *zxIs170[pmyo-3::ChR2, pmyo-3::RCaMP35]*, **ZX3305:** *unc-31(n1304) IV; egl-19(nu698[EGL-19 mCherry11 knock-in]) IV; nuTi335[ppat-10::sfCherry1-10]*, **ZX3314:** *unc-31(n1304) IV; zxEx1534[punc-17β::unc-31::SL2:: mCherry+ punc-47::unc-31::SL2::mCherry]; oxIs353*, **ZX3322:** *unc-31(n1304) IV; zxIs170*, **ZX3333:** *unc-31(n1304) IV; zxEx1533[punc-47::unc-31::SL2::mCherry]; oxIs353*, **ZX3338:** *npr-18(ok1388) X; oxIs353*, **ZX3373:** *kpc-1(gk8) I; oxIs353*, **ZX3374:** *bli-4(e937) I; oxIs353*, **ZX3534:** *aex-5(sa23) I; zxIs170*, **ZX3535:** *egl-3(gk238) V; zxIs170*, **ZX3536:** *aex-5(sa23) I; egl-3(gk238) V; zxIs170*, **ZX3553:** *aex-5(sa23) I; zxEx1445[paex-5::aex-5::SL2::GFP]; zxIs170*, **ZX3594:** *unc-31(n1304) IV; zxEx1528[pmyo-3::unc-31::SL2::GFP]; oxIs353*, **ZX3595:** *unc-31(n1304) IV; zxEx1529[prab-3::unc-31::SL2::GFP]; oxIs353*, **ZX3787:** *ace-1(ok663) X; ace-2(ok2545) I; oxIs353*, **ZX3788:** *ace-1(ok663) X; ace-2(ok2545) I; unc-31 (n1304) IV; oxIs353*, **ZX3854:** *aex-5(sa23) I; zxEx1511[punc-47::aex-5::P2A::GFP]; zxIs170*, **ZX3855:** *zxEx1512 [pmyo-3::egl-19b::P2A::GFP]; zxIs170*, **ZX3896:** *nlp-9(tm3572) V; zxEx1520[punc-17::nlp-9::SL2::mCherry]; oxIs353*, **ZX3897:** *nlp-9(tm3572) V; zxEx1521[punc-47::nlp-9::SL2::mCherry]; oxIs353*, **ZX3922:** *nlp-9(tm3572) V; npr-18(ok1388) X; oxIs353*, **ZX3923:** *aex-5(sa23) I; zxEx1539[punc-17::aex-5::P2::GFP]; zxIs170*, **ZX3924:** *flp-9(ok2730) IV; oxIs353*, **ZX3925:** *flp-15(gk1186) III; oxIs353*, **ZX3926:** *nlp-15(ok1512) I; oxIs353*, **ZX3927:** *unc-31(n1304) IV; zxEx1537[pacr-2:: nlp-9::mCherry; pmyo-2::CFP]*, **ZX4002:** *nlp-9(tm3572) V; nlp-21(tm2569) III; nlp-38(ok2330) I; zxIs53*, **ZX4003:** *egl-3(gk238)V; aex-5(sa23) I; zxIs53*, **ZX4004:** *flp-15(gk1186) III; zxIs53*, **ZX4005:** *nlp-15(ok1512) I; zxIs53*, **ZX4006:** *flp-15(gk1186) III; nlp-15(ok1512) I; zxIs53*, **ZX4007:** *unc-31(e928) IV; zxIs53*, **ZX4008:** *unc-31(e928)IV; oxIs353*, **ZX4009:** *krSi234[Punc-17::TIR1::BFP::unc-54 3′ UTR]; rp166(unc-31-linker-GFP-TEV- AID-FLAG) IV; oxIs353*, **ZX4010:** *nlp-9(tm3572)V; egl-19(nu698[EGL-19 mCherry11 knock-in]) IV; nuTi335[ppat-10::sfCherry1-10]*, **ZX4011:** *unc-31(n1304) IV; egl-19(nu674[EGL-19 GFP11 knock-in]); nuSi285[punc-47::GFP1-10::SL2::UNC-57::mCherry]*, **ZX4012:** *unc-31(n1304) IV; egl-19(nu674[EGL-19 GFP11 knock-in]); nuSi250[punc-129::GFP1-10::SL2::UNC-57::mCherry]*, **ZX4035:** *egl-3(gk238) V; zxEx1558[punc-17::egl-3::P2A::gfp, pmyo-2::mCherry]; zxIs170*, **ZX4036:** *egl-3(gk238) V; zxEx1559[punc-47::egl-3::P2A::gfp, pmyo-2::mCherry]; zxIs170*, **ZX4040:** *cha-1(p1152) IV; oxIs353*

## Molecular biology

Plasmids generated or used in this work are listed in S2 Table, and primers used are listed in S3 Table.

## Rescue constructs

*unc-31* cDNA was amplified by PCR from KG#121, a gift from Kenneth Miller (Addgene plasmid # 110879; http://n2t.net/addgene:110879; RRID: Addgene_110879), and inserted using HiFi assembly (NEB) into expression vectors containing

the *rab-3* promoter (pan-neuronal rescue), the *unc-17* promoter (cholinergic rescue), the *unc-47* promoter (GABAergic rescue), and the *myo-3* promoter (muscle rescue), respectively.

*egl-3* gDNA was amplified by PCR from pJH2124, a gift from Mei Zhen (Addgene plasmid # 178791; http://n2t.net/addgene:178791; RRID: Addgene_178791), and inserted into expression vectors containing the *unc-17* promoter, and the *unc-47* promoter respectively.

*aex-5, nlp-9,* and *nlp-38* genomic sequence were amplified by PCR from N2 genomic DNA and inserted into expression vectors respectively.

Genomic fragments as promoters, upstream of the start codon of *aex-5* (2 kb), *nlp-9* (2.8 kb), and *nlp-38* (2 kb), were amplified by PCR from N2 genomic DNA.

## *egl-19* RNAi

A 1.5 kb genomic fragment of *egl-19* [65] was amplified from N2 genomic DNA and inserted into plasmid L4440 by restriction digest with *HindIII* and *NheI*. The construct was transformed in *E. coli* strain HT115 [72]. dsRNA induction was performed as previously described [73]. Bacteria were grown over night on LB-Ampicillin-containing agar dishes. Colonies were picked and cultivated over night at 37 °C and 180 rpm in 5 ml LB-Ampicillin supplemented media. NGM Ampicillin plates were seeded with 200 µl culture/plate and incubated at 37 °C over night. Next day, *egl-19* RNAi feeding bacteria were induced to synthesize dsRNA by application of 150 µl Isopropyl-β-ᴅ-thiogalacto-pyranosid (IPTG) stock solution (100 mM), and then 15 L4 animals were transferred onto those plates. A second batch of NGM-Ampicillin plates was seeded with the *egl-19* RNAi feeding strain. These were induced on the following day by 150 µl (100 mM) IPTG stock containing all-trans-retinal (ATR; 4 µl stock solution of 100 mM), and 10 adult animals from the respective first culture dish were transferred onto the new culture dish and their progeny cultivated for 4 days.

## Auxin inducible degron (AID) assays

The strain for cholinergic neuron-specific depletion of UNC-31 using the AID was generated by crossing an N-terminal GFP-AID-FLAG-UNC-31 knock-in strain (RJP5269, [27], kindly provided by Roger Pocock) with a strain expressing TIR1 in cholinergic neurons (EN8636, kindly provided by Florence Solari; [74]).

Auxin- or ethanol-containing NGM plates were prepared as previously described [27]. For consistent UNC-31 depletion, L4 animals were cultivated overnight on auxin- or ethanol-NGM plates and transferred to fresh auxin- or ethanol-NGM plates. Animals from the next generation were used for measurements. For acute UNC-31 depletion, day 1 adult animals were cultivated on auxin- or ethanol-NGM plates for 5 h prior to experiments.

## *egl-19* overexpression

*egl-19b* cDNA was amplified by PCR from KP#2460 (kindly provided by Joshua M. Kaplan). The cDNA was inserted into expression vector that contains the muscular expression promoter *myo-3*.

## Transgenes and germline transformation

Transgenic strains were generated by microinjection of various DNA constructs with a co-injection marker (pmyo-2::mCherry (2 ng/µl), pmyo-2::CFP (2 ng/µl), or pttx-3::mScarlet (30 ng/µl). Integrated strains were obtained by histone-miniSOG-induced mutagenesis [75], followed by outcrossing against N2 at least 4 times.

## Behavioral assays

Crawling speed assays were performed in a multiworm tracker (MWT) platform [33], illuminated by LED modules ALUSTAR (Ledxon GmbH, Geisenhausen, Germany) at 70 µW/mm², and as previously described [76,77]. Due to the high

sensitivity of bPAC, all experiments dealing with bPAC-expressing strains were performed in a dark room, and animals were manipulated under red-filtered light [21]. An infrared back light on the MWT was made with WEPIR3-S1 IR Power LED Star infrared (850 nm) 3W (Winger Electronics GmbH & Co. KG, Dessau, Germany) powered by LCM-40 constant current LED driver and regulated with a potentiometer (Vishay Intertechnology, Inc., Malvern, PA, USA) to avoid bPAC pre-activation. To synchronize worms, around 15 gravid animals were transferred to OP50-1-seeded NGM plates and allowed to lay eggs for 6−8 h, and then removed. After 3−4 days, around 80 day-1 adults were collected using M9 buffer and transferred onto plain NGM plates for measurements.

Body contraction assays were performed in a Zeiss Axiovert 40 microscope (Zeiss, Germany), illuminated with blue light from a 50 W HBO mercury lamp filtered through a GFP excitation filter (450−490 nm) at 1.1, 0.1, and 65 µW/mm$^2$ intensities as indicated in respective figure legends, under a 10× objective. L4 larvae were transferred to freshly seeded ATR plates (seeded with 300 µl OP50-1 culture, mixed with 0.6 µl of 100 mM ATR stock dissolved in ethanol) and single day-1 adults were placed onto plain NGM plates for measurements. The body length was determined as previously described [78].

Animals were kept in darkness for 15 min before all recordings. The measurements were repeated on 2–3 different experimental days.

## Fluorescence imaging

For calcium imaging, combined with optogenetic stimulation, RCaMP35- [34] and ChR2-expressing animals were mounted on glass slides and immobilized with polystyrene beads (0.1 µm diameter, at 2.5% w/v, Sigma-Aldrich). Images were taken with a Zeiss Axio Observer Z1 compound microscope equipped with a Kinetix22 (Teledyne Photometrics, Tucson, AZ, USA) camera, Prior Lumen LEDs (Prior Scientific, Cambridge, UK), and an RCaMP filter cube (a dual-band excitation filter 479/585 nm, a 647 nm emission filter, and a 605 nm bean splitter, F74-423, F37-647, and F38-605, respectively; AHF Analysentechnik, Germany), with a 20× objective with illumination light at 470 nm to activate ChR2 for defined periods, with excitation light at 590 nm for RCaMP signal recording.

Voltage imaging was performed as previously described [55]. Briefly, QuasAr-expressing animals were immobilized with polystyrene beads and imaged on a Zeiss Axio Observer Z1 microscope equipped with 40× oil immersion objective (Zeiss EC Plan-NEOFLUAR ×40/N.A. 1.3, Oil DIC ∞/0.17), a laser beam splitter (HC BS R594 lambda/2 PV flat, AHF Analysentechnik), and a galilean beam expander (BE02-05-A, Thorlabs). Voltage-dependent fluorescence of QuasAr was excited with a 637 nm laser (OBIS FP 637LX, Coherent) at 1.8 W/mm$^2$ and imaged at 700 nm, while ChR2 was activated by a monochromator (Polychrome V) at 300 µW/mm$^2$.

The coelomocyte assay was performed as previously described [48]. Briefly, NLP-9::mCherry fusion protein-expressing animals were mounted on glass slides and immobilized with tetramisole. Images were taken with Zeiss Observer Z1 with 40× objective with 590 nm illumination with RFP filter cube (ex. 580/23 nm, em. 625/15 nm). All coelomocytes within each animal were imaged for quantification. For bPAC strains, worms were illuminated with blue light at 35 µW/mm$^2$ for 15 min prior to imaging.

Images of representative expression patterns of *nlp-9 and nlp-38* transcriptional reporter constructs, as well as *EGL-19::mCherry* images for quantification, were acquired with a Zeiss LSM780 microscope with 63× oil objective (Plan-Apochromat 63×/1.4 Oil DIC) at 488 and 543 nm for GFP and mCherry excitation, respectively.

Image were subjected to background subtraction and quantification conducted with Fiji [79].

## Electrophysiology

Electrophysiological recordings of body wall muscle cells were done in dissected adult worms as previously described [18]. Animals were immobilized with Histoacryl L glue (B. Braun Surgical, Spain) and a lateral incision was made to access NMJs along the anterior ventral nerve cord. The basement membrane overlying body wall muscles was enzymatically

removed by 0.5mg/ml collagenase for 10 s (C5138, Sigma-Aldrich, Germany). Integrity of BWMs and nerve cord was visually examined via DIC microscopy. Recordings from BWMs were acquired in whole-cell patch-clamp mode at 20−22 °C using an EPC-10 amplifier equipped with Patchmaster software (HEKA, Germany). The head stage was connected to a standard HEKA pipette holder for fire-polished borosilicate pipettes (1B100F-4, Worcester Polytechnic Institute, USA) of 4–10 MΩ resistance. The extracellular bath solution (CRG) consisted of 150 mM NaCl, 5 mM KCl, 5 mM $CaCl_2$, 1 mM $MgCl_2$, 10 mM glucose, 5 mM sucrose, and 15 mM HEPES, pH 7.3, with NaOH, ~330 mOsm. The internal patch pipette solution consisted of K-gluconate 115 mM, KCl 25 mM, $CaCl_2$ 0.1 mM, $MgCl_2$ 5 mM, BAPTA 1 mM, HEPES 10 mM, $Na_2ATP$ 5 mM, $Na_2GTP$ 0.5 mM, cAMP 0.5 mM, and cGMP 0.5 mM, pH 7.2, with KOH, ~320 mOsm.

Voltage clamp experiments were conducted at a holding potential of −60 mV. Light activation was performed using an LED lamp (KSL-70, Rapp OptoElectronic, Hamburg, Germany; 470 nm, 8 mW/mm$^2$) and controlled by the Patchmaster software. Puff-application (80 ms) of ACh (500 μM in CRG) to record ACh-evoked currents were performed using a Pico-spritzer III (Parker, USA). Subsequent analysis and graphing was performed using Patchmaster, and Origin (Originlabs). Analysis of mPSCs was conducted with MiniAnalysis (Synaptosoft, Decatur, GA, USA, version 6.0.7). Membrane potential and APs in BWM cells were recorded in current clamp mode. To induce APs, an additional current pulse of +20 pA (50 ms) was injected at 10.01 s via the Patchmaster software for 50 ms. For data analysis, voltage traces were aligned to the peak of the AP, and replotted, thus appear shifted relative to the current pulse.

## Statistical analysis

All quantitative data are shown as mean ± s.e.m. *n* indicates the number of animals, and *N* indicates biological replicates/measurement sessions with distinct populations of animals, unless otherwise noted. Significance between data sets after two-tailed Student *t* test (two groups comparison), one-way AVOVA (multiple groups comparison), with Bonferroni's or Tukey's multiple comparison test, or Fisher test, or two-way ANOVA (multiple groups comparison), were performed as indicated in each figure legend. Data were analyzed and plotted in GraphPad Prism (GraphPad Software, version 8.02) or OriginPro 2024 (64-bit; SR1 10.1.0.178; Origin Lab Corporation).

## Supporting information

**S1 Fig. Related to** Fig 1. **(A, B)** Quantification of bPAC::YFP signal in ventral nerve cord cholinergic neurons of wt ($n = 28$) and *unc-31* mutants ($n = 19$). Representative images (A; Scale bar 10 μm) and summary data (B) are shown. **(C, D)** Cholinergic ChR2 activation-evoked postsynaptic currents in wt ($n = 7$) and *unc-31* mutants ($n = 7$). Representative traces (C) and group data of peak currents (D) are shown. **(E, F)** Quantification of ChR2::mCherry signal in body wall muscles of wt ($n = 48$) and *unc-31* mutants ($n = 34$). Representative images (E; Scale bar 10 μm) and summary data (F) are shown. **(G, H)** Quantification of GFP-AID-UNC-31 intensity in nerve ring after ethanol ($n = 15$) or auxin ($n = 15$) 5 h exposure compared to control animals ($n = 16$, no treatment). Representative images (G; Scale bar 10 μm) and summary data (H) are shown. **(I, J)** Measurements of body length induced by muscular ChR2 activation after UNC-31 depletion using 0.1 mW/mm$^2$ blue light stimulation. Auxin or ethanol treated animals were compared. Animal number tested in each group: $n = 12$, 13 respectively. Blue bar indicates the 5–15 s blue light illumination. The data in panel J represent the mean values over the entire illumination period shown in panel I. Data presented as mean ± SEM. Statistical significance for two-group datasets and multiple-group datasets comparison was determined using unpaired t test and one-way ANOVA with Tukey-correction respectively. ** and **** indicate $p < 0.01$ and $p < 0.0001$, respectively. Numerical data can be found in S1 Dataset.
(TIF)

**S2 Fig. Related to** Fig 2. **(A, B)** Measurements of body length change, induced by muscular ChR2 activation using 65 μW/mm$^2$ blue light stimulation. *aex-5* genomic rescue was done by specifically expressing AEX-5 under its own promoter.

Animal number tested $n = 46$, 24, 22, from left to right, respectively. The data in panel B represent the mean values over the entire illumination period (5–15 s) shown in panel A. **(C)** AEX-5 is released from GABAergic neurons. Coelomocyte fluorescence, resulting from expression in, using the *unc-47* promoter, and secretion of mCherry-tagged AEX-5 from GABAergic neurons. White arrows indicate the coelomocytes in the middle and anterior parts of the animal. White arrowheads indicate the ventral cord GABAergic neurons. Scale bar 100 μm. Data presented as mean ± SEM. Statistical significance was determined using one-way ANOVA with Tukey-correction. ** and **** indicate $p < 0.01$ and $p < 0.0001$, respectively. Numerical data can be found in S1 Dataset.
(TIF)

**S3 Fig. Related to Fig 3. (A)** Expression data summary of neuropeptides expressed in MNs. Heatmap representing the expression patterns and levels of neuropeptide mRNAs in different neuron types were generated from CeNGENApp (https://cengen.shinyapps.io/CengenApp/). The initial exported plot was cropped to show cholinergic neurons, GABAergic neurons, and the cholinergic interneuron DVA. Scaled TPM (transcripts per million) and proportion data refer to all neurons of *Caenorhabditis elegans*, not only the subset shown. **(B)** Mean speed traces following bPAC stimulation in cholinergic neurons, of the animals indicated in, and giving rise to group data of Fig 3C and 3D.
(TIF)

**S4 Fig. Related to Fig 4. (A, B)** Measurements of body length induced by muscular ChR2 activation using 100 μW/mm² blue light stimulation in wt ($n = 11$), *nlp-9; nlp-21; nlp-38* ($n = 14$), *nlp-9* ($n = 13$), and *unc-31* mutants ($n = 9$). The data in panel B represent the mean values over the entire illumination period (5–10 s) shown in panel A. **(C–F)** Crawling speed induced by cholinergic bPAC activation was compared in the indicated genotypes. *nlp-9* (C, D) and *nlp-38* rescue (E, F) were done by specifically expressing NLP-9 and NLP-38 from the cholinergic promoter *unc-17*. Animal number tested in (D) $n = 60$–80, in $N = 12$, 16, 8 experiments, and in (F) $n = 60$–80, in $N = 8$, 6, 6 experiments, from left to right, respectively. All data are presented as mean ± SEM. Statistical significance for two-group datasets and multiple-group datasets comparison was determined using unpaired t test and one-way ANOVA with Tukey-correction respectively. *** and **** indicate $p < 0.001$ and $p < 0.0001$, respectively. Numerical data can be found in S1 Dataset.
(TIF)

**S5 Fig. Related to Fig 4. (A, B)** Anatomical expression patterns of *nlp-9* and *nlp-38*. Representative images of *nlp-9* and *nlp-38* transcriptional reporters driving mCherry fluorescence from the *nlp-9* and *nlp-38* promoters, respectively and their colocalization with the fluorescence of YFP, expressed in cholinergic neurons using the *unc-17* promoter. White arrowheads indicate colocalization of cell bodies of *nlp-9* or *nlp-38* expressing neurons and ventral cord cholinergic MNs. Scale bars 100 and 20 μm, respectively.
(TIF)

**S6 Fig. Related to Fig 6. (A–F)** Data as in Fig 6B–6G, but not normalized. Analysis of (m)PSC rate and amplitudes in body wall muscle cells, induced by bPAC stimulation of cholinergic neurons. Mean ± SEM mPSCs, recorded from dissected body wall muscle of adult worms in wt ($n = 19$), *aex-5; egl-3* ($n = 7$), *nlp-9* ($n = 13$), *nlp-38* ($n = 8$), *nlp-9; nlp-21; nlp-38* ($n = 7$), and *flp-15; nlp-15* ($n = 7$) mutants. Blue bar indicates stimulation of bPAC in cholinergic neurons. Note, the same wt control data was used in panels A, C, E and B, D, F.
(TIF)

**S7 Fig. Related to Fig 7. (A, B)** Basal, unstimulated mPSCs recorded from dissected BWM cells of adult worms in wt ($n = 8$) and *unc-31* mutants ($n = 9$). Representative traces of mPSCs (A) and summary data of mPSC amplitudes (B) are shown. **(C, D)** Quantification of cholinergic ChR2::GFP intensity in ventral nerve cord of wt ($n = 31$) and *unc-31* ($n = 24$) mutants. Representative images (C; Scale bar 10 μm) and summary data (D) are shown. **(E)** Voltage imaging using the fluorescent voltage indicator QuasAr, expressed in BWM cells of wt ($n = 13$) and *unc-31* mutants ($n = 12$),

during depolarization evoked by ChR2 stimulation of cholinergic neurons, as in Fig 7G and 7H. The first peak $\Delta F/F_0$ is the initial maximal signal observed during ChR2 stimulation. All data are presented as mean ± SEM. Statistical significance comparison was determined using unpaired t test. ns = not significant. Numerical data can be found in S1 Dataset.
(TIF)

**S8 Fig. Related to** Fig 8**. (A, B)** Increased evoked muscle contraction in *aex-5* mutant is reverted by CaV1 block. Body contraction induced by muscular ChR2 activation using 65 μW/mm² blue light stimulation was compared in the indicated genotypes and treatments. Relative body lengths after treatment with the CaV1 specific inhibitor nemadipine are shown. Number of animals tested, $n = 14, 14, 14, 15$, from left to right columns, respectively. The data in panel B represent the mean values over the entire illumination period (5–15 s) shown in panel A. Data are presented as mean ± SEM. Statistical significance comparison was determined using two-way ANOVA with Sidak's multiple comparisons test. * and **** indicate $p < 0.05$ and $p < 0.0001$, respectively. Numerical data can be found in S1 Dataset.
(TIF)

**S9 Fig. Related to** Fig 9**. (A, B)** Endogenous expression levels of EGL-19 in muscle cells were compared in wt ($n = 14$) and *nlp-9* mutants ($n = 14$). Representative images (A; Scale bar 20 μm) and summary data (B) are shown. **(C–F)** Expression of endogenously GFP-tagged EGL-19 in cholinergic (C, D) and GABAergic neurons (E, F) were quantified in wt ($n = 28$ and 22, respectively) and *unc-31* mutants ($n = 29$ and 25, respectively). Representative images (C, E; Scale bar 20 μm) and summary data (D, F) are shown. **(G–I)** Enhanced AP amplitude in *unc-31* mutants; delayed and prolonged APs in animals overexpressing EGL-19 CaV1. Individual (gray) and mean (red) traces of membrane potential changes recorded from BWM cells after 20 pA current-step induced at 10.01 s, in wt (G), *unc-31* (H), and EGL-19 over-expressing (OEx) animals (I) are shown, from $n = 15, 15$, and 14 animals, respectively. Note that the traces were aligned to the time of the peak amplitude, and replotted, thus they appear shifted to earlier times. All data are presented as mean ± SEM. Statistically significant comparisons were determined using unpaired t test. *** indicates $p < 0.001$. Numerical data can be found in S1 Dataset.
(TIF)

**S1 Table. Sequences of neuropeptide precursors studied in this work, and putative cleavage sites for AEX-5 and EGL-3 pro-protein convertases.**
(DOCX)

**S2 Table. Plasmids generated in this work.**
(DOCX)

**S3 Table. Primers used in this work.**
(DOCX)

**S1 Data. Numerical data related to** Fig 1**.**
(XLSX)

**S2 Data. Numerical data related to** Fig 2**.**
(XLSX)

**S3 Data. Numerical data related to** Fig 3**.**
(XLSX)

**S4 Data. Numerical data related to** Fig 4**.**
(XLSX)

**S5 Data. Numerical data related to** Fig 5.
(XLSX)

**S6 Data. Numerical data related to** Fig 6.
(XLSX)

**S7 Data. Numerical data related to** Fig 7.
(XLSX)

**S8 Data. Numerical data related to** Fig 8.
(XLSX)

**S9 Data. Numerical data related to** Fig 9.
(XLSX)

**S1 Dataset. Numerical data related to** S1–S9 Figs.
(XLSX)

## Acknowledgments

We would like to express our gratitude to members of the Gottschalk lab for supporting this research. We are particularly indebted to Franziska Baumbach and Katharina Kuhlmeier for excellent technical support, and to Ichiro Aoki for critically reading and commenting on the manuscript. We thank Kenneth Miller, Roger Pocock, Florence Solari, Stephen Nurrish, and Joshua Kaplan for reagents and strains. Some strains were obtained from the *Caenorhabditis* Genetics Center (CGC), which is funded by the NIH Office of Research Infrastructure Programs (P40 OD010440). Other strains were obtained from the Japanese National Bioresource Project for the nematode, funded by the Japan Agency for Medical Research and Development (AMED).

## Author contributions

**Conceptualization:** Jiajie Shao, Wagner Steuer Costa, Alexander Gottschalk.

**Data curation:** Jiajie Shao, Jana F. Liewald, Alexander Gottschalk.

**Formal analysis:** Jiajie Shao, Jana F. Liewald, Wagner Steuer Costa, Alexander Gottschalk.

**Funding acquisition:** Alexander Gottschalk.

**Investigation:** Jiajie Shao, Jana F. Liewald, Wagner Steuer Costa, Christiane Ruse, Jens Gruber, Mohammad S. Djamshedzad, Wulf Gebhardt.

**Methodology:** Jiajie Shao, Jana F. Liewald.

**Project administration:** Alexander Gottschalk.

**Resources:** Jiajie Shao, Wagner Steuer Costa.

**Supervision:** Alexander Gottschalk.

**Validation:** Jiajie Shao, Jana F. Liewald, Alexander Gottschalk.

**Visualization:** Jiajie Shao, Jana F. Liewald, Alexander Gottschalk.

**Writing – original draft:** Jiajie Shao, Alexander Gottschalk.

**Writing – review & editing:** Jiajie Shao, Jana F. Liewald, Alexander Gottschalk.

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
