## [Editor Report · Decision Letter 0]

24 Sep 2024

Dear Dr Gottschalk, 

Thank you for submitting your manuscript entitled "Loss of neuropeptidergic regulation of cholinergic transmission induces CaV1-mediated homeostatic compensation in muscle cells" for consideration as a Research Article by PLOS Biology.

Your manuscript has now been evaluated by the PLOS Biology editorial staff as well as by an academic editor with relevant expertise and I am writing to let you know that we would like to send your submission out for external peer review.

Once your full submission is complete, your paper will undergo a series of checks in preparation for peer review. After your manuscript has passed the checks it will be sent out for review. To provide the metadata for your submission, please Login to Editorial Manager (https://www.editorialmanager.com/pbiology) within two working days, i.e. by Sep 26 2024 11:59PM.

Kind regards,

Christian

Christian Schnell, PhD

Senior Editor

PLOS Biology

cschnell@plos.org

---

## [Decision Letter · Decision Letter 1]

12 Nov 2024

Dear Alexander,

Thank you for your patience while your manuscript "Loss of neuropeptidergic regulation of cholinergic transmission induces CaV1-mediated homeostatic compensation in muscle cells" was peer-reviewed at PLOS Biology. It has now been evaluated by the PLOS Biology editors, an Academic Editor with relevant expertise, and by several independent reviewers. 

In light of the reviews, which you will find at the end of this email, we would like to invite you to revise the work to thoroughly address the reviewers' reports.

As you will see below, the reviewers think that the study is overall well executed and provides important insights. However, they all raise concerns, for example about the degree of variability, the interpretability, the role of muscle compensation, lack of controls and other models that could potentially explain the findings.

Given the extent of revision needed, we cannot make a decision about publication until we have seen the revised manuscript and your response to the reviewers' comments. Your revised manuscript is likely to be sent for further evaluation by all or a subset of the reviewers.

**IMPORTANT - SUBMITTING YOUR REVISION**

*Re-submission Checklist*

*Published Peer Review*

*PLOS Data Policy*

*Blot and Gel Data Policy*

Sincerely,

Christian

Christian Schnell, PhD

Senior Editor

PLOS Biology

cschnell@plos.org

REVIEWS:

Reviewer #1 (Shangbang Gao): This manuscript offers an innovative exploration of how neuropeptidergic regulation impacts cholinergic transmission and the resulting compensatory mechanisms in muscle cells in C. elegans. The authors present a novel finding that reduced acetylcholine output due to impaired presynaptic neuropeptide signaling triggers homeostatic upregulation of the L-type voltage-gated calcium channel EGL-19 in postsynaptic muscle cells, effectively preserving synaptic strength. This discovery adds a new layer to our understanding of synaptic plasticity and highlights the intricate balance between pre- and post-synaptic mechanisms in maintaining neuromuscular function. Overall, this study provides significant insights and a valuable contribution to the field of neurobiology.

Major Comments:

1. The proposed mechanism of postsynaptic homeostatic scaling, mediated by increased muscle excitability via upregulation of L-VGCC/EGL-19, is a key and intriguing finding. This compensatory mechanism response to reduced ACh-output in neuropeptide signaling mutants is central to this study. However, an important question arises: is this a general principle across different contexts of reduced cholinergic transmission? For instance, would similar postsynaptic compensation occur in other mutants with impaired cholinergic transmission? I was curious to ask if this phenomenon is also present in mutants such as unc-13, unc-17, or unc-2 etc. ? Additional experiments addressing this question would greatly enhance our understanding of whether the observed EGL-19 upregulation is solely due to reduced ACh-output or if other contributing factors co-exist. 

2. The authors mention that egl-19 is expressed in both presynaptic neurons and postsynaptic muscle cells. My second question: Is egl-19 also upregulated in presynaptic neurons in the unc-31 mutants and other neuropeptide mutants identified in this study? While this question may seem outside the primary scope of this study, addressing this is crucial for a complete understanding of the proposed conclusions. If egl-19 is similarly altered in presynaptic neurons, it could indicate that neuropeptides directly regulate egl-19 expression in multiple tissues. This would help clarify that the behavioral effects observed in the mutants are not solely due to increased egl-19 expression in muscle cells. Surely, it is possible that neuropeptides use distinct mechanisms to regulate egl-19 in nerve and muscle cells. If so, wouldn't that further expand the depth of research in this paper!

3. The use of the Ach::bPAC optogenetic method to demonstrate reduced mPSCs in nlp-9 and nlp-38 mutants is compelling. However, several points warrant clarification: Do unc-31, egl-3, and axe-5 mutants exhibit similar declines in mPSCs? What was the rationale behind using the Ach::bPAC optogenetic method for this experiment? Are endogenous mPSCs and EPSCs (evoked via ACh::ChR2 stimulation) similarly reduced in mutants such as egl-3, axe-5, nlp-9, nlp-38, FLP-15, and NLP-15? These clarifications would be critical to validate the reduction in ACh-output in the mutants analyzed.

4. The draft provided by the author lacks all supplementary figures, some of the data cannot be judged substantively.

Minor Concerns:

1. It would be beneficial to examine whether the phenomenon of postsynaptic compensation observed in unc-31(n1304) can be consistently reproduced in different unc-31 mutant alleles. This would help to solidify the findings.

2. In Fig. 2, egl-3 mutants exhibit enhanced muscle excitability even with milder stimulation. Could the authors elaborate on the potential mechanism driving this phenomenon? Additionally, a more detailed description of the types of neurons in which egl-3 primarily functions is needed.

3. In Fig. 5B,C, Ach::bPAC activation results in a modest increase in NLP-9 secretion. Could the authors provide further explanation as to why this increase is weak? Is it possibly related to the duration of light stimulation?

4. In Fig. 5DE, does the nlp-9; nlp-38 double mutant show a further reduction in mPSCs? Additionally, does unc-31(lf) also reduce Ach::bPAC-excited mPSCs?

5. In Fig. 8A, do other neuropeptide mutants exhibit upregulated endogenous levels of muscular EGL-19, similar to what is observed in unc-31(lf)? Investigating this would help establish whether this compensatory mechanism is generalizable across other mutants.

6. In Fig. 8E, the action potentials do not fully resemble the standard waveforms previously recorded in similar studies. This discrepancy may have affected the analysis and contributed to inconsistencies in the text when comparing unc-31(lf) and EGL-19 OEx effects. It may be more informative to directly record voltage-gated calcium currents in WT, unc-31(lf), and EGL-19 OEx strains to assess whether functional calcium currents are indeed increased.

Overall, this study presents several innovative and important findings on the role of neuropeptidergic regulation in synaptic plasticity. Addressing these points would further strengthen the conclusions and provide deeper insight into the mechanisms underlying cholinergic transmission and compensatory regulation in C. elegans muscle cells.

Reviewer #2: In this manuscript, Shao et al. seek to determine whether neuropeptidergic signaling modulates synaptic transmission through homeostatic plasticity, specifically via the upregulation of postsynaptic CaV1 channels. They use optogenetic bPAC stimulation in cholinergic motor neurons to induce cAMP-dependent neuropeptide release, coupled with behavioral experiments highlighting the mutant phenotype of unc-31 (CAPS ortholog, responsible for dense core vesicle release). Rescue experiments indicate that a complex neuromodulatory interface exists between cholinergic and GABAergic neuropeptides at the neuromuscular junction (NMJ). The identification of AEX-5 and EGL-3 as key proprotein convertases that work jointly in UNC-31-dependent signaling at the NMJ is a noteworthy contribution. The study provides evidence that cholinergically expressed neuropeptides NLP-9 and NLP-38 modulate locomotion and muscle contraction. In addition, FLP-15 and NLP-15, highly expressed in GABAergic neurons, also function to modulate muscle contractions. Overall, they find a compensatory enhancement in postsynaptic Ca2+ influx following the loss of UNC-31 using optogenetics, calcium imaging, and voltage imaging. They identify EGL-19, an L-type VGCC, as a critical player in this muscle homeostasis. While some experiments are hare to interpret, and there is a high degree of variability in many of the assays, overall, the findings are novel, and the following adjustments will make the contributions clearer to readers.

Major comments

1. Overall the high degree of variability in the phenotypes of even control strains from experiment to experiment makes the data a bit hard to interpret. For example, wt body length upon muscle ChR2 stimulation is around 0.95 in figure 1H, but closer to 0.98 in Fig. S3, where triple mutants are now at 0.95 (and concluded to be defective in their excitability). If the experimental variability between experimental sessions is as great as the variability between wt and mutant, how can the data be properly interpreted? A bit more discussion of what might be leading to this variability is necessary.

In addition, how summary data were calculated is not clear (i.e.- how figure 1H was generated from figure 1G… is that an average over the entire time? It is the value at a particular time point?). 

2. Absence of unc-31 mutant controls in Figures 3D-E and 4C-H limits the interpretability of the data regarding neuropeptidergic modulation of muscle cell excitability. It would be informative to know how these individual peptide phenotypes compare to the unc-31 mutant phenotype. Moreover, the double (nlp-9; nlp-38) or triple mutants (nlp-9; nlp-21; nlp-38) mutant data should be presented alongside single mutant data (rather than separately in supplemental material) so they can be directly compared.

3. The methods for the experiments shown in Figures 5D-H are not adequately described, specifically what is being expressed and where (ChR2? bPAC?) and what time points are the summary data in G-H being drawn from? From panels E-F, it seems like the mutants are still responding initially to the stimulation, albeit more weakly, but that the response is not as sustained as in wt. However, the summary data suggest that the mutants have no change at all in mPSC amplitude or frequency upon stimulation. This is confusing.

4. The minimal effect observed in wild-type animals with both nemadipine and especially RNAi treatments is unexpected, given the central role of EGL-19 in muscle excitability. This suggests that only the "compensatory" channels are affected, a point that, if believed true, should be elaborated on in the discussion. The timing of the RNAi application is not clearly stated- is it added late in development or are the worms grown up on the RNAi plates? Could this play a role in the lack of effect in wt? Also, the specificity of the RNAi to the muscle EGL-19 (due to the lack of neuronal sensitivity to RNAi) should be mentioned.

Minor comments

1. The unc-31 mutant phenotype involves an emergence of a latent secondary peak in speed approximately 100 seconds after photostimulation, accompanied by an extended decay. The authors may consider elaborating on this phenomenon in the text to provide further insight.

2. The study aimed to identify neuropeptides responsible for modulating cholinergic signaling by extracting expression pattern information from previously published data. However, the rationale for the selection of candidate neuropeptides from the CenGen scRNAseq database remains unclear. Also NLP-12 is said to have previously been studied, but neither context nor citation is included.

3. A distinction should be made between when something is "significantly" increased or decreased (which means there is a large difference) versus when there are small but "statistically significant" differences. At several points in the text this distinction is lost, for example, lines 378 and 540, where a small (albeit statistically significant) difference is described as being "significant" (ie- large). 

4. In figure 5E-H, the Y-axis is labeled "mPSC," even though blue light is being used to evoke a response, the response is clearly larger than individual quantal events, and the text on line 383 refers to them as "evoked" postsynaptic currents. A clearer distinction between (or definition of) mPSC and evoked PSC is needed in both the figures and the manuscript text.

5. Inconsistency in stimulation duration (ranging from 10 to 40 seconds) across optogenetics assays needs justification. Standardizing this protocol would enhance the comparability of results.

6. The novelty and impact of the manuscript would shine with revisions for clarity and succinctness by simplifying complex sentences and removing any redundancy.

7. The final figure summarizing the proposed model would greatly benefit from a more descriptive title and perhaps a simpler layout. For example, removing the DVA neuron, which was not discussed in detail, and using a distinct symbol for autocrine release would improve clarity. A subcellular view of the model, in addition to the current circuit view, might better capture the detailed mechanisms explored in the manuscript.

Reviewer #3: This is an interesting study from Gottschalk and colleagues that aims to dissect the interplay of pre- versus postsynaptic impacts of neuropeptide (NP) signaling at the C. elegans NMJ. The work builds from a prior 2017 publication from the same lab that provides evidence for cholinergic presynaptic modulation through neuropeptide regulation of vesicular loading. The current study extends these findings to explore locomotory effects of NP modulation, identify specific neuropeptides involved, and define a potential pathway for postsynaptic homeostatic compensation through upregulation of EGL-19 L-type voltage-gated calcium channels in the muscles of NP-deficient mutants. The overall idea is intriguing and the experimental results presented largely support the authors' model; however, several of the assays are indirect or employ different methodologies across experiments, in some cases making it difficult to draw strong conclusions. As just one example, the authors often switch between presynaptic bPAC (cAMP) stimulation for the behavioral studies and ChR2 stimulation for the electrophysiology studies, making it very difficult to relate the 2 sets of findings. There are a number of interesting results here, but alternative models that are not deeply explored could also account for many of the findings. I provide detailed comments below.

1. Figures 1A,E: It would be interesting to show whether ChR2 stimulation of cholinergic motor neurons elicits the same increases. If the effects are mediated through postsynaptic compensation due to loss of NP signaling, I would predict a similar behavioral response and increase in muscle calcium for ChR2 stimulation of motor neurons in unc-31 mutants.

2. For many of the experiments, the authors show rescue with ACh-specific unc-31 expression. While this shows sufficiency, experiments aimed at showing necessity in ACh motor neurons would strengthen the analysis considerably. Using a reagent such as the available unc-31::AID strain for this purpose would also provide temporal control for demonstrating post-developmental roles, for instance in the case of EGL-19 upregulation in unc-31 mutants (Figure 8). 

3. In figure 3, different sets of NP-deficient mutants are surveyed across the 2 behavioral assays. It would be more informative if all of the mutants were included for each assay, as this would provide evidence for specificity of the effects.

4. Different experimental paradigms are used across many of the figures, making comparisons challenging and weakening the strength of some of the conclusions. For example, including recordings of evoked current responses to ChR2 photostimulation of motor neurons in Figure 6 would allow for direct comparison with the ACh-evoked response in panels 6H-I, and enhance the impact of the lack of an effect on the ACh-evoked response. Also, inclusion of a current clamp recording of muscle APs in response to ChR2 stimulation of motor neurons would strengthen the argument that a reduced current elicits increased levels of muscle excitation in unc-31 mutants. Finally, a voltage clamp recording from muscles showing increased calcium current density would be the most effective way to demonstrate upregulation of EGL-19. The increase in AP amplitude in Figure 8E could arise through a variety of mechanisms. I am not suggesting that that authors need to perform all of these experiments, but instead pointing out that that the indirect nature of some of the approaches used and the accompanying limitations in interpretation should be noted more explicitly.

5. Since the authors have not defined cognate receptors and their expression, it is difficult to rule out the possibility that changes in muscle excitability could arise due to direct effects of NPs on muscle. In particular, the effects of NPs thought to be released from GABA motor neurons are confusing. It seems the authors are suggesting GABA neuron-released peptides are required to sustain ACh transmission by acting to promote the filling of cholinergic SVs. While interesting, additional work is required to better support this model.

6. Mechanisms that link autocrine effects of NPs with SV filling are left unexplored. Likewise, potential mechanisms for EGL-19 upregulation are not explored. These could easily be topics for future standalone papers, but are nonetheless areas where there is a lack of clarity in the present study. Collectively, this leaves the reader with the impression that the results are mostly considered in the context of a pre-existing framework with less attention to other possibilities.

Minor comments

1. The authors should provide evidence that bPAC and/or ChR2 expression is not impacted in the various mutants, at a minimum for unc-31.

2. It would be more clear if rescue strains were denoted with (+). For example, in Figure 2E: aex-5;ACh::aex-5(+)

3. Line 283 and elsewhere: The authors could perhaps be a little more cautious in relating the CenGen scRNA-seq data to protein expression. While the CenGen data are undoubtedly an incredible resource, they are unlikely to completely capture native protein expression across cell types.

4. Line 291: "GABAergic FLP-15" is confusing. If it is also in other neurons, it is not exclusively GABAergic. The authors should rephrase.

5. Did the authors examine impacts of flp-15;nlp-15 double mutants since the single mutants each elicit partial effects compared to unc-31 (Figure 3D,E versus Fig 2A-D). Also, it looks as if the same data for unc-31 may be used for comparison across panels 2A and 2C. If this is the case, it should be noted in the figure legend.

6. Line 335: this experiment measures muscle contraction which the authors interpret as a proxy for muscle excitability. Stating the measurement as "muscle contraction" would be more clear.

7. Lines 355-356: cell type specific rescue shows sufficiency, not a requirement.

8. Figure 5D-H: It is not stated whether these recordings are measuring responses to ChR2 or bPAC stimulation. I interpret the data as ChR2 stimulation. As stated above, it is difficult to relate the bPAC-elicited behavioral responses and the Chr2-elicited electrophysiology responses. In figure 5G, it is unclear what statistical comparison the single asterisk refers to.

9. The most appropriate citation for characterization of CKR-1 as an NLP-12 receptor is Ramachandran et al elife, 2021 as cited in the Beets 2023 paper.

---

## [Decision Letter · Decision Letter 2]

3 Apr 2025

Dear Alexander,

Thank you for your patience while we considered your revised manuscript "Loss of neuropeptidergic regulation of cholinergic transmission induces CaV1-mediated homeostatic compensation in muscle cells" for publication as a Research Article at PLOS Biology. This revised version of your manuscript has been evaluated by the PLOS Biology editors, the Academic Editor and the original reviewers.

Based on the reviews and on our Academic Editor's assessment of your revision, we are likely to accept this manuscript for publication, provided you satisfactorily address the remaining points raised by the reviewers. Please also make sure to address the following data and other policy-related requests:

* We would like to suggest a different title to improve its accessibility for our broad audience: 

Loss of neuropeptidergic regulation of cholinergic transmission induces homeostatic compensation in muscle cells to preserve synaptic strength

* Please add the links to the funding agencies in the Financial Disclosure statement in the manuscript details.

* DATA POLICY:

Regardless of the method selected, please ensure that you provide the individual numerical values that underlie the summary data displayed in the following figure panels as they are essential for readers to assess your analysis and to reproduce it: 1BDFHJL, 2BDFH, 3CD, 4BDFHJ, 5C, 6HI, 7BDFHJLN, 8BDF, 9BDFGHI, S1BDFH, S2B, S4BDF, S7BDE, S8B and S9BDF.

* CODE POLICY

* Please note that per journal policy, we do not allow the mention of "data not shown", "personal communication", "manuscript in preparation" or other references to data that is not publicly available or contained within this manuscript. Please either remove mention of these data or provide figures presenting the results and the data underlying the figure(s).

We expect to receive your revised manuscript within two weeks. 

*Published Peer Review History*

*Press*

Sincerely,

Christian

Christian Schnell, PhD

Senior Editor

cschnell@plos.org

PLOS Biology

Reviewer remarks:

Reviewer #1 (Shangbang Gao): This paper represents a significant contribution to the field, offering valuable insights for assessing compensatory neural functional changes. The authors have adequately addressed my concerns and has supplemented the necessary experiments or provided additional explanations. I have no further questions.

Reviewer #2: A couple of points from the first review were not adequately addressed.

Relevant to previous Major Comment 1: The rationale for varying stimulation frequency was explained in the rebuttal, but we could not find it in the manuscript itself. The manuscript would benefit from a clear explanation for how and why stimulation frequencies were chosen for the various genotypes used. Summary data were not labeled as being an average over the entire stimulation time in the figures or figure legends in the revised manuscript, even though the rebuttal states that it was. This should be added to the figure legends.

Relevant to previous Major Comment 3: Figure 6 now more clearly labels that the methods used in Figure 6A are bPAC expression in cholinergic neurons. However, the X-axes in Figure 6H and I confusingly label different aspects of the experiment (light stimulation and genotypes). Both panels should have both sets of labels as there is space available to do so.

Additional comment: In Figure 4, relative body length recordings and respective quantifications do not match. Figure 4D belongs next to Figure 4G, Figure 4H belongs next to Figure 4I, and Figure 4J belongs next to Figure 4C. This likely oversight should be corrected to avoid confusion. 

Reviewer #3: The authors have addressed most of my prior comments. I have only a few suggestions for additional clarification.

Lines 78-79: I would suggest the authors amend this sentence to state, " A major mode of action of these peptides…." or similar, as it is unlikely that modulation of SV filling is their exclusive mode of action.

Line 96: missing a "we"

Line 244: I think the authors are referring to 2C here instead of 1C.

---

## [Editor Report · Decision Letter 3]

17 Apr 2025

Dear Alexander,

Thank you for the submission of your revised Research Article "Loss of neuropeptidergic regulation of cholinergic transmission induces homeostatic compensation in muscle cells to preserve synaptic strength" for publication in PLOS Biology. On behalf of my colleagues and the Academic Editor, Mark Alkema, I am pleased to say that we can in principle accept your manuscript for publication, provided you address any remaining formatting and reporting issues. These will be detailed in an email you should receive within 2-3 business days from our colleagues in the journal operations team; no action is required from you until then. Please note that we will not be able to formally accept your manuscript and schedule it for publication until you have completed any requested changes.

PRESS

Sincerely, 

Christian

Christian Schnell, PhD

Senior Editor

PLOS Biology

cschnell@plos.org